# Wolf2Pack: The AutoFusion Framework for Dynamic Parameter Fusion

## Abstract

In the rapidly evolving field of deep learning, specialized models have driven significant advancements in tasks such as computer vision and natural language processing. However, this specialization leads to a fragmented ecosystem where models lack the adaptability for broader applications. To overcome this, we introduce **AutoFusion**, an innovative framework that fusing distinct model's parameters(with the same architecture) for multi-task learning without pre-trained checkpoints. Using an unsupervised, end-to-end approach, AutoFusion dynamically permutes model parameters at each layer, optimizing the combination through a loss-minimization process that does not require labeled data. We validate AutoFusion's effectiveness through experiments on commonly used benchmark datasets, demonstrating superior performance over established methods like Weight Interpolation, Git Re-Basin, and ZipIt. Our framework offers a scalable and flexible solution for model integration, positioning it as a powerful tool for future research and practical applications.

> "For the strength of the pack is the wolf,
> and the strength of the wolf is the pack."

*- Rudyard Kipling*

## 1 Introduction

In the rapidly evolving landscape of technological innovation, deep learning models have become increasingly specialized Dong et al. (2021) Lai et al. (2024a) Li et al. (2024) Lai et al. (2022) Lai et al. (2024b), leading to substantial advancements in diverse fields such as computer vision and natural language processing Sharifani & Amini (2023). These specialized models, meticulously honed to excel in their designated niches, have undeniably propelled numerous breakthroughs Taye (2023). However, this specialization has inadvertently led to a fragmented ecosystem where models, although highly effective within their specific domains, lack the adaptability and versatility required to address a wider array of challenges. This raises a pivotal question: Is it possible to amalgamate the strengths of these specialized models into a unified architecture capable of performing multiple tasks proficiently?

The challenge of integrating specialized models into a coherent system is multifaceted. Traditional approaches to model fusion heavily depend on prior knowledge and require meticulous tuning of hyperparameters, such as specifying which layers to merge and permuting parameters according to what principle. Ainsworth et al. (2022) Stoica et al. (2023) Qu & Horvath (2024). Do we have to propose a new approach to any new problem? This is obviously costly and has a serious impact on the usefulness of parametric fusion. Moreover, In the case where model parameters do not share pre-trained parameters, merging parameters from different tasks obviously cannot be directly accomplished through the previously common method of parameter permutation based on similarity.

To address these challenges, we propose **AutoFusion**, an innovative framework designed to **fuse the parameters of two models, which do not share the same pre-trained parameters and perform different tasks, into a single parameter capable of simultaneously accomplishing multiple tasks** which can be expressed figuratively as Figure 1. Drawing inspiration from the principle that 'more is merrier', unlike conventional methods that depend on predefined rules or heuristics, **AutoFusion aims to learn an effective permutation of model parameters to accomplish the fusing of multi-task model parameters.** This unsupervised training process requires no labeled data, making it a flexible and scalable solution for model integration.

Figure 1: In this figure we use deer and sheep to denote different tasks, and different colors of wolves to denote different models, our purpose is to make a reasonable fusion of models that are good at each, which can make the fusion model good at different tasks.

The AutoFusion method is primarily based on two operations: aligning parameters that perform similar functions through permutation, and retaining parameters that perform different functions through their permutation as much as possible. The key to achieving this is the design of the loss function, which guides the learning process. Specifically, the loss function is designed to minimize the discrepancy between the fused model's output and the outputs of the individual specialized models on their respective tasks. This ensures that the fused model retains the strengths of the original models while being able to generalize across multiple tasks.

The unsupervised nature of AutoFusion is a critical aspect of its design. By not requiring labeled data, AutoFusion can be applied to a wide range of model architectures and datasets, making it a versatile tool for future research and practical applications in deep learning. The end-to-end design of AutoFusion allows for dynamic permuting of model parameters at each layer, resulting in a unified and robust model capable of handling multiple tasks.

To evaluate the efficacy of AutoFusion, we conducted experiments on the commonly used benchmark datasets Xiao et al. (2017) Clanuwat et al. (2018) LeCun et al. (1998) Krizhevsky et al. (2009), simulating the scenario of merging models trained on distinct tasks. Our findings indicate that the merged model achieves high accuracy across all sub-tasks, frequently outperforming established techniques like Weight Interpolation Li et al. (2023), Git Re-Basin Ainsworth et al. (2022), and ZipIt Stoica et al. (2023).

Our contributions to the field of model parameter integration can be summarized as follows:

**(i) End-to-End Unsupervised Framework:** We present an end-to-end, unsupervised approach to model parameter fusion, eliminating the need for prior knowledge and predefined hyperparameters. This approach facilitates the dynamic permuting of model parameters at each layer, resulting in a unified and robust model capable of handling multiple tasks.

**(ii) Empirical Validation and Performance:** Through extensive experimentation on commonly used benchmark datasets, we demonstrate the superior performance of AutoFusion compared to established methods such as Weight Interpolation, Git Re-Basin, and ZipIt. Our framework achieves high accuracy across all sub-tasks, highlighting its effectiveness in multi-task scenarios.

**(iii) Scalability and Flexibility:** The unsupervised nature of AutoFusion ensures its scalability and flexibility, allowing it to be applied to a wide range of model architectures and datasets without the need for labeled data. This characteristic positions our framework as a versatile tool for future research and practical applications in deep learning.

AutoFusion represents a significant stride forward in the field of model parameter fusion. By offering an end-to-end, unsupervised approach to model fusion, we aim to unify the disparate threads of specialized deep-learning models into a cohesive, adaptable ecosystem. This work not only advances the state-of-the-art in model fusion but also paves the way for future research into more versatile and efficient deep-learning architectures. Our code will be available on GitHub after the explanation of the double-blind review.

## 2 PRELIMINARY

The main problem addressed in our work can be defined as follows: in the absence of shared pre-trained parameters (without sharing the optimization process), we aim to fuse the parameters of two identical architecture models trained separately on disjoint tasks to obtain a fused model Stoica et al. (2023). The expectation is that this fused model can retain, to the greatest extent possible, the capabilities of each model before fusion.

If two datasets of disjoint tasks are recorded as datasets A and B:

$$\mathcal{D}_i = \{(x_j, y_j)|j \in N^i\}, i \in \{A, B\} \tag{1}$$

where $N^i$ indicates the number of samples in the dataset $\mathcal{D}_i$. Suppose the cross-entropy loss can be expressed as $\mathcal{H}(\cdot)$, then the models trained on datasets A and B can be represented as $\Theta_i, i \in \{A, B\}$, where $\Theta_i$ is derived from the following formula:

$$\underset{\Theta_i}{argmin} \frac{1}{N^i} \sum_{j=0}^{N^i} \mathcal{H}(P_m(x_j|\Theta_i), y_j) \tag{2}$$

$P_m(\cdot)$ represents the predicted output of input $x_j$ based on the model parameter $\Theta_i$. when we get the parameters of the two models, $\Theta_A$ and $\Theta_B$. Next, let's assume that the model's parametric fusion operation can be represented as:

$$\Theta_{merged} = \mathcal{M}(\Theta_A, \Theta_B) \tag{3}$$

Our goal is to find an $\mathcal{M}$ that minimizes the joint loss of the fused model $\Theta_{merged}$ on datasets A and B, which can be expressed as:

$$\underset{\mathcal{M}}{argmin} \frac{1}{2} \sum_{i=A}^{\{A,B\}} \frac{1}{N^i} \sum_{j=0}^{N^i} \mathcal{H}(P_m(x_j^i|\Theta_{merged}), y_j^i) \tag{4}$$

### 2.1 WEIGHT INTERPOLATION

Early parametric fusion relied primarily on the ability to perform arithmetic averaging directly to the model's parameters to integrate the model Frankle et al. (2020) Wortsman et al. (2022b) Matena & Raffel (2022) Wortsman et al. (2022a) Izmailov et al. (2018), a process that can be represented as:

$$\mathcal{M}(\Theta_A, \Theta_B, \gamma) = \gamma\Theta_A + (1-\gamma)\Theta_B = \{\gamma W_A^l + (1-\gamma)W_B^l|l \in [0, L]\} \tag{5}$$

where $L$ denotes the number of layers of the model, and the parameters $W$ of each layer are treated as a vector in space $\mathbb{R}^{d_l}$, and $\gamma$ always set to $\frac{1}{2}$.

However, this method is quite crude. When the two models do not share common pre-trained parameters, their parameters often cannot be directly corresponded due to the permutation invariance of neural networks, making it difficult to obtain valuable results through direct linear interpolation Singh & Jaggi (2020) Ainsworth et al. (2022) Neyshabur et al. (2020) Gao et al. (2022).

### 2.2 RE-BASIN

Addressing the issues arising from directly applying linear interpolation to parameters, studies Xiao & Cheng (2023) Entezari et al. (2021) Peyré et al. (2019) propose that since randomly permuting neurons within a neural network does not affect the final output, we can first align the parameters of two models by permutation. This involves corresponding neurons responsible for the same functions to the same positions in both models before performing linear interpolation. This process can be represented as follows:

$$\mathcal{M}(\Theta_A, \Theta_B, \gamma) = \gamma\Theta_A + (1-\gamma)\pi(\Theta_B) = \{\gamma W_A^l + (1-\gamma)P_l W_B^l P_{l-1}^T|l \in [0, L]\} \tag{6}$$

where $\pi(\cdot)$ represents the transformation using the corresponding permutation matrix $P$ for each layer, $P_l \in \pi$ represents the permutation matrix of layer $l$, and to eliminate the influence of the layer $l-1$ permutation on the current layer, it is also multiplied by the inverse matrix of the layer $l-1$ permutation matrix $P_{l-1}^{-1}$, but since the permutation matrix is orthogonal, its inverse matrix is equal to its transpose matrix $P_{l-1}^T$. The permutation matrix $P$ is solved using the layer-by-layer greedy linear assignment method(Hungarian Algorithm). The goal of the method optimization can be expressed as:

$$\underset{\pi}{argmin}\, d(\Theta_A, \pi(\Theta_B)) \tag{7}$$

where $d(\cdot)$ denotes the distance between the two model parameters, which can be further expressed as:

$$d(\Theta_A, \pi(\Theta_B)) = \frac{1}{L} \sum_{l=0}^{L-1} \| W_A^l - P_l W_B^l P_{l-1}^T \|^2 \tag{8}$$

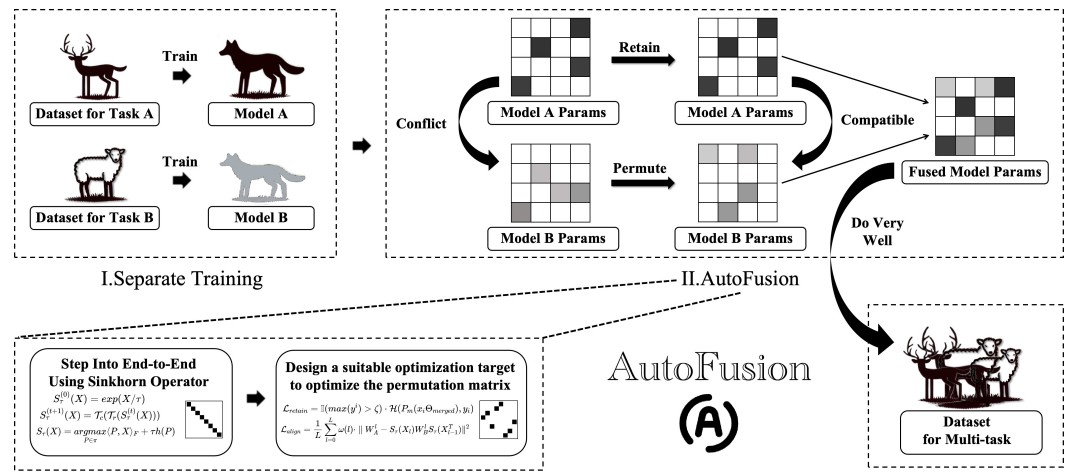

Figure 2: This is an overview of our AutoFusion methodology, implementation details can be found in section 3

This method can align neurons with similar functions to a certain extent, allowing for the integration of parameters from two models through linear interpolation without losing accuracy. However, such an operation tends to make the parameters of the two models similar, making it unsuitable for scenarios where different models need to retain their diversity when merging multi-task models.

### 2.3 MODEL ZIP

In the context of the aforementioned research, Zipit Stoica et al. (2023), for the first time, proposed a method targeting the issue of multi-task parameter fusion without pre-trained parameters. This method considers the activation values of each layer's output in the model, employing a merging matrix($M$) to combine features with high correlation while utilizing an unmerging matrix($U$) to reverse the merging when features cannot be effectively combined.

If we express the activation value of layer l as $f_l$, Then we can express the above operation as:

$$f_l^* = M_l(f_l^A \parallel f_l^B), \; U_l f_l^* \simeq f_l^A \parallel f_l^B \tag{9}$$

where $\parallel$ stands for combination operation. Unlike previous work, Zipit takes into account the self-matching of activation values.

After getting $U$ and $M$ matrices, Zipit uses these matrices to transform the parameters and fuse the parameters:

$$W_l^* = M_l^A W_A^l U_{l-1}^A + M_l^B W_B^l U_{l-1}^B \tag{10}$$

Although Zipit's model compression method has improved the effectiveness of multi-task model fusion to some extent, it remains confined to merging similar functionalities through parameter permutation. The approach adopted by Zipit, which enhances fusion by forsaking the merging of layers with weaker similarities, does not genuinely address the underlying issues of multi-task merging but rather serves as a compromise solution out of necessity. Therefore, it is evident that exploring model parameter fusion methods under complete merging scenarios remains imperative.

## 3 AUTOFUSION

AutoFusion proposes a novel parameter fusion method to address the issue of multi-task model parameter fusion in the absence of pre-trained parameters(These models do not share the same pre-training weights and are all trained from a random initialization). An overview of the AutoFusion method is shown in Figure 2. The specific design methodology is detailed in the following subsections.

### 3.1 FROM RULE-BASED TO END-TO-END

Existing methods primarily rely on manually designed rules for parameter alignment, which are limited by the assumption that parameters of the same layer should exhibit high similarity. How-

ever, this assumption of high similarity falls apart when the models to be merged are trained for different tasks. During merging, we must not only align parameters with similar functions but also strive to retain parameters with distinct functions, enabling the fused model to perform various tasks simultaneously.

Determining which parameters with different functions to retain is a challenge that cannot be easily addressed through prior knowledge Stoica et al. (2023). It cannot be achieved through simple similarity metrics and straightforward rules, as is the case with parameter similarity alignment. This compels us to consider advancing towards an end-to-end approach, where model parameter fusion is accomplished directly through learning.

We attempted to utilize neural functional functions from neural functional analysis to predict network parameters from network parameters Navon et al. (2023) Zhou et al. (2024b) Zhou et al. (2024a). Specifically, employing permutation-invariant neural networks to directly accept network parameters as input and output the fused network parameters:

$$vec(\Theta^*) = \Psi(vec(\Theta^A), vec(\Theta^B)) \tag{11}$$

where $vec(\cdot)$ represents flattening the parameter to a high-dimensional vector, $\Psi(\cdot)$ denotes the neural function used for fusion. However, the excessively large number of parameters in this scheme results in high training costs, making it challenging for practical application. Considering that the rows and columns of the model parameter matrix inherently contain complete information, and the cost required to learn the permutation matrix is minimal, this naturally leads us to shift our focus towards learning the parameter permutation matrix.

Inspired by Mena et al. (2018) and Peña et al. (2023), we employ the Sinkhorn operator to convert the discrete permutation matrix into a differentiable form to satisfy the criteria for gradient descent optimization, first defining:

$$S_\tau(X) = \underset{P \in \pi}{argmax} \langle P, X \rangle_F + \tau h(P) \tag{12}$$

where $\langle A, B \rangle_F$ represents $trace(A^T B)$, and $\pi$ represents the set of all permutation matrices that have the same shape as $X$, $X$ is a $N$ dimensional square matrix. $h(P)$ represents the entropy regularizer $-\sum_{i,j} P_{i,j} log P_{i,j}$, and $\tau$ represents it's weight.

Equation 12 is known as the Sinkhorn operator, the matching operation of the permutation matrix is not differentiable, but Mena et al. (2018) proves that a differentiable computational step can approximate it:

$$\begin{aligned} S_\tau^{(0)}(X) &= exp(X/\tau) \\ S_\tau^{(t+1)}(X) &= \mathcal{T}_c(\mathcal{T}_r(S_\tau^{(t)}(X))) \end{aligned} \tag{13}$$

where $X \in \mathbb{R}^{n \times n}$, it can be seen as a soft version of the permutation matrix, $\mathcal{T}_r(X)$ and $\mathcal{T}_c(X)$ represent the operation of normalizing the rows and columns of the matrix, respectively, can be calculated as $X \oslash (X \mathbf{1}_n \mathbf{1}_n^T)$ and $X \oslash (\mathbf{1}_n \mathbf{1}_n^T X)$ where $\oslash$ stands for element-wise division. Mena et al. (2018) proves that when $t \to \infty$, Equation 13 converges to Equation 12.

To prove the credibility of this approximation, we derive the upper error bound between it and the Sinkhorn operator:

**Theorem (Error Bound for the Sinkhorn Operator)**: For any fixed $\tau > 0$, the approximation error $\mathcal{E}$ satisfies the following inequality:

$$\mathcal{E} \leq \frac{\|X\|_\infty^2}{2\tau} \tag{14}$$

The detailed proof process and differentiability of approximate calculations are given in Appendix C.

This then allows us to learn the appropriate parameter permutation matrix by setting up the appropriate loss function, and the process of merging can be expressed as:

$$\begin{aligned} \mathcal{M}_{AF}(\Theta_A, \Theta_B, \gamma) &= \gamma \Theta_A + (1 - \gamma) \pi(\Theta_B) \\ &= \{\gamma W_A^l + (1 - \gamma) S_\tau(X_l) W_B^l S_\tau(X_{l-1}^T) | l \in [0, L]\} \end{aligned} \tag{15}$$

## 3.2 Design of Optimization Targets

We have now constructed a learnable permutation matrix using the Sinkhorn operator, which can be directly applied to parameter fusion. Therefore, the next step is to design a reasonable optimization objective to refine the permutation matrix, enabling the fused model to integrate parameters for common functionalities while preserving the necessary parameter diversity for handling multiple tasks.

According to the conclusions of Yosinski et al. (2014) Taye (2023) Zhou et al. (2022), some representations learned by neural networks tend to have strong generality, and the generality representations can often be merged through parameter alignment to improve the stability of the network for these representations Ainsworth et al. (2022).

To align these neurons, we designed a weighted parametric alignment loss:

$$\mathcal{L}_{align} = \frac{1}{L} \sum_{l=0}^{L} \omega(l) \cdot \parallel W_A^l - S_\tau(X_l) W_B^l S_\tau(X_{l-1}^T) \parallel^2 \tag{16}$$

where $\omega(l)$ represents the loss weight of the current layer $l$. The reason for performing layer-wise weighting is that most studies have shown that features learned by shallow-layer neurons tend to be more generalizable. In AutoFusion, we chose to set $\omega(l)$ for each layer by linear relationship $\omega(l) = \frac{2L}{l}$.

To encourage the permutation matrix learned by the model to retain features that can handle multiple tasks to a certain extent, we randomly sampled a batch of input data from multi-task dataset $\mathcal{D}_{sampled} = \{x_i | i \in [0, N_s)\}$. Our goal is to leverage accessible model parameters to obtain reliable pseudo-labels for this data without accessing their true labels, thereby assisting in the training of the permutation matrix. Firstly, we utilized existing models A and B to obtain their predictions for this data:

$$\widehat{Y}_k = \{y_k^i = P_m(x_i | \Theta_k) | x_i \in \mathcal{D}_{sampled}\}, k \in \{A, B\} \tag{17}$$

Next, we define $\mathbb{C}(\cdot)$ that can choose the one with higher confidence from the prediction output of the two models as the final output:

$$\mathbb{C}(y_A, y_B) = \mathbb{I}(\max(y_A) > \max(y_B)) \cdot y_A + \mathbb{I}(\max(y_B) > \max(y_A)) \cdot y_B \tag{18}$$

where $\mathbb{I}(\cdot)$ is the indicator function, with a value of 1 when the conditions are met and a value of 0 when the conditions are not met. Next, we can use $\mathbb{C}(\cdot)$ to complete the screening of the output of the two models:

$$\widehat{Y} = \{y^i = \mathbb{C}(y_A^i, y_B^i) | y_A^i \in \widehat{Y}_A, y_B^i \in \widehat{Y}_B\} \tag{19}$$

After obtaining $\widehat{Y}$, it is necessary to construct a computational graph containing the parameters of the permutation matrix to be trained through operations, to complete supervised learning. Inspired by Peña et al. (2023), we sample a fusion coefficient $\gamma_t$ from a uniform distribution represented as $\gamma_t \sim U(0, 1)$ and fuse the models to be combined using the existing permutation matrix following the method of Equation 15:

$$\Theta_{merged} = \mathcal{M}_{AF}(\Theta_A, \Theta_B, \gamma_t) \tag{20}$$

Next, the permutation matrix optimization goal that retains multitasking capabilities can be expressed as:

$$\mathcal{L}_{retain} = \mathbb{I}(max(y^i) > \zeta) \cdot \mathcal{H}(P_m(x_i \Theta_{merged}), y_i) \tag{21}$$

where, $y_i \in \widehat{Y}$, paired one-to-one with $x_i$, and $x_i$ is the input sample from $\mathcal{D}_{sampled}$ and $\zeta$ is a hyperparameter that represents the selected confidence threshold to filter low confidence predictions. Now we can optimize the permutation matrix by combining $\mathcal{L}_{align}$ and $\mathcal{L}_{retain}$ together for training:

$$\mathcal{L} = \omega_a \cdot \mathcal{L}_{align} + \omega_r \cdot \mathcal{L}_{retain} \tag{22}$$

where $\omega_a$ and $\omega_r$ are the weights of $\mathcal{L}_{align}$ and $\mathcal{L}_{retain}$, respectively. It is important to note that during the whole training process, only the permutation matrix of the parameters will be trained, and the parameters of any model will not save the gradient.

## 4 RESULTS

Due to the scarcity of work on multi-task model parameter fusion without pretraining, we have partially adopted the settings from Stoica et al. (2023) in designing our experiments. In Table 4.1, we split several commonly used benchmark datasets in computer vision into non-overlapping subsets based on their categories, trained models on these subsets independently, and compared the effects of parameter fusion using different methods and different network structures. We have also included crucial ablation experiments subsection 4.2 and parameter experiments in Appendix E to comprehensively evaluate the method's effectiveness and parameter sensitivity. In subsection 4.4, we present some visualization results to demonstrate the model's effectiveness from a more intuitive perspective.

### 4.1 COMPARISON WITH OTHER METHODS

| Dataset | Method | Joint | TaskA | TaskB | Avg |
|---|---|---|---|---|---|
| MINIST(5+5) MLP | Model A | $58.92 \pm 0.01$ | $97.26 \pm 0.01$ | $19.42 \pm 0.01$ | 58.34 |
| | Model B | $53.00 \pm 0.01$ | $9.45 \pm 0.01$ | $97.84 \pm 0.01$ | 53.65 |
| | Ensemble Model | $97.12 \pm 0.5$ | $96.98 \pm 0.3$ | $97.12 \pm 0.7$ | 97.05 |
| | Weight Interpolation | $53.06 \pm 0.01$ | $67.99 \pm 0.01$ | $37.67 \pm 0.01$ | 52.83 |
| | Git Re-Basin | $50.08 \pm 0.4$ | $45.12 \pm 1.1$ | $52.99 \pm 1.0$ | 49.06 |
| | Zipit | $51.25 \pm 0.6$ | $57.31 \pm 1.2$ | $45.00 \pm 0.7$ | 51.25 |
| | AutoFusion | $\mathbf{85.85} \pm 0.7$ | $\mathbf{88.56} \pm 0.8$ | $\mathbf{83.04} \pm 0.8$ | $\mathbf{85.80}$ |
| CIFAR-10(5+5) MLP | Model A | $45.16 \pm 0.01$ | $62.30 \pm 0.01$ | $28.02 \pm 0.01$ | 45.16 |
| | Model B | $43.83 \pm 0.01$ | $24.01 \pm 0.01$ | $63.56 \pm 0.01$ | 43.83 |
| | Ensemble Model | $59.23 \pm 0.9$ | $58.12 \pm 1.1$ | $60.23 \pm 1.2$ | 59.18 |
| | Weight Interpolation | $20.01 \pm 0.01$ | $20.00 \pm 0.01$ | $20.02 \pm 0.01$ | 20.01 |
| | Git Re-Basin | $40.12 \pm 0.3$ | $37.13 \pm 0.4$ | $\mathbf{44.01} \pm 0.2$ | 40.57 |
| | Zipit | $40.58 \pm 0.2$ | $38.48 \pm 0.3$ | $42.68 \pm 0.2$ | 40.58 |
| | AutoFusion | $\mathbf{45.10} \pm 0.1$ | $\mathbf{47.47} \pm 0.1$ | $42.76 \pm 0.2$ | $\mathbf{45.12}$ |
| MINIST(5+5) CNN | Model A | $57.11 \pm 0.01$ | $97.85 \pm 0.01$ | $10.39 \pm 0.01$ | 54.12 |
| | Model B | $54.35 \pm 0.01$ | $17.24 \pm 0.01$ | $98.86 \pm 0.01$ | 58.05 |
| | Ensemble Model | $98.13 \pm 2.2$ | $97.63 \pm 1.7$ | $98.22 \pm 1.9$ | 97.93 |
| | Weight Interpolation | $21.15 \pm 0.01$ | $22.34 \pm 0.01$ | $19.89 \pm 0.01$ | 21.12 |
| | Git Re-Basin | $52.08 \pm 1.1$ | $19.15 \pm 1.8$ | $85.99 \pm 1.0$ | 52.57 |
| | Zipit | $52.00 \pm 0.6$ | $50.19 \pm 1.2$ | $52.31 \pm 0.7$ | 51.25 |
| | AutoFusion | $\mathbf{65.23} \pm 0.2$ | $\mathbf{58.65} \pm 0.3$ | $\mathbf{72.58} \pm 0.2$ | $\mathbf{65.62}$ |
| CIFAR-10(5+5) CNN | Model A | $45.69 \pm 0.01$ | $81.34 \pm 0.01$ | $26.01 \pm 0.01$ | 53.67 |
| | Model B | $44.31 \pm 0.01$ | $23.86 \pm 0.01$ | $83.66 \pm 0.01$ | 53.67 |
| | Ensemble Model | $79.11 \pm 3.1$ | $79.73 \pm 2.7$ | $78.21 \pm 2.2$ | 78.97 |
| | Weight Interpolation | $20.01 \pm 0.01$ | $20.05 \pm 0.01$ | $20.11 \pm 0.01$ | 20.08 |
| | Git Re-Basin | $39.41 \pm 0.3$ | $30.32 \pm 0.4$ | $45.15 \pm 0.5$ | 37.73 |
| | Zipit | $47.65 \pm 0.4$ | $48.78 \pm 1.2$ | $45.99 \pm 1.3$ | 47.38 |
| | AutoFusion | $\mathbf{52.85} \pm 0.7$ | $\mathbf{53.24} \pm 0.5$ | $\mathbf{52.46} \pm 0.6$ | $\mathbf{52.85}$ |

Table 1: AutoFusion test results on different feature extraction networks and different datasets.

**Baselines** To assess the superiority of AutoFusion, we selected several widely used methods in the field of parameter fusion as our comparison objects, namely Weight Interpolation, Git Re-BasinAinsworth et al. (2022), and ZipitStoica et al. (2023). Among them, Git Re-Basin represents the most widely used solution for mainstream parameter alignment methods, and after testing, we only chose the Weights Matching method, which yielded the best results. Zipit, on the other hand, is the first method specifically designed to address multi-task parameter fusion without pretraining. Weight Interpolation is the most straightforward method in parameter fusion. We also included data from directly evaluating the unfused model to highlight the effectiveness of the parameter fusion methods.

**Datasets** We selected two commonly used benchmark datasets in the field of computer vision, MNIST and CIFAR-10, both of which are 10-class datasets. Using random sampling, we split these 10-class datasets into two non-overlapping 5-class datasets, denoted as Dataset A and Dataset B, following the settings in section 2. Subsequently, we independently trained models on the divided datasets to obtain the multi-task models ready for fusion.

**Settings** To comprehensively evaluate the performance of AutoFusion under different architectures, we selected MLP and CNN (VGG)Simonyan & Zisserman (2015) as the base networks for eval-

uation. We independently trained models on different model architectures and different parts of datasets. We used various fusion methods for parameter fusion and analyzed the accuracy of the fused models. The **"Joint"** column represents the accuracy of the current model tested on the undivided dataset, while **"Task A"** and **"Task B"** represent the accuracy of the model tested on Dataset A and Dataset B respectively. **"Avg"** simply denotes the arithmetic average of the results from Task A and Task B. **Model A(B)** indicates a model that has been trained only on Dataset A(B). More specific parameter settings are provided in subsection D.2.

**Analysis** Our main experimental results are presented in subsection 4.1. The data in these tables represent accuracy rates, and the standard deviations of the data are calculated based on five consolidation operations after a single model training session. Both the Git Re-Basin[1] method and the Zipit[2] method utilize officially released codes for model fusion. Observing these results, we can find there is a significant improvement in joint accuracy using AutoFusion. When evaluating the Fused CNN model on the MNIST dataset, AutoFusion surpassed Zipit, the previously most advanced model, achieving a 13.23% improvement in joint accuracy. And for the Fused MLP Model, AutoFusion almost outperformed Zipit's results by 34.6%. Correspondingly, the AutoFusion method has been greatly improved on both Task A and Task B. The results on CIFAR-10 show that although the improvement on this dataset is not as large as that of the MINIST dataset, it still maintains the SOTA in joint accuracy.

## 4.2 ABLATION STUDY AND OPTIMIZATION STRATEGIES

| Model | Method | Joint | TaskA | TaskB | Avg |
|-------|--------|-------|-------|-------|-----|
| CNN | Model A | $57.11 \pm 0.01$ | $97.85 \pm 0.01$ | $10.39 \pm 0.01$ | 54.12 |
| | Model B | $54.35 \pm 0.01$ | $17.24 \pm 0.01$ | $98.86 \pm 0.01$ | 58.05 |
| | Weight Interpolation | $25.44 \pm 0.01$ | $18.58 \pm 0.01$ | $32.50 \pm 0.01$ | 25.54 |
| | Weighted Optimize | $61.12 \pm 1.1$ | $51.51 \pm 0.8$ | $71.01 \pm 0.9$ | 61.26 |
| | Rounded Optimize | $62.33 \pm 0.1$ | $52.90 \pm 1.8$ | $72.15 \pm 1.2$ | 62.53 |
| | Normalized Optimize | $\mathbf{65.23} \pm 0.2$ | $\mathbf{58.65} \pm 0.3$ | $\mathbf{72.58} \pm 0.2$ | **65.62** |
| | $\mathcal{L}_{align}$ Only | $36.00 \pm 1.3$ | $21.58 \pm 2.9$ | $50.85 \pm 2.0$ | 36.22 |
| | $\mathcal{L}_{retain}$ Only | $60.98 \pm 1.3$ | $53.92 \pm 1.2$ | $68.25 \pm 1.2$ | 61.08 |
| MLP | Model A | $58.71 \pm 0.01$ | $96.57 \pm 0.01$ | $19.71 \pm 0.01$ | 58.14 |
| | Model B | $52.86 \pm 0.01$ | $9.89 \pm 0.01$ | $97.12 \pm 0.01$ | 53.51 |
| | Weight Interpolation | $33.76 \pm 0.01$ | $40.08 \pm 0.01$ | $27.24 \pm 0.01$ | 33.66 |
| | Weighted Optimize | $82.10 \pm 0.4$ | $86.12 \pm 0.3$ | $77.95 \pm 0.8$ | 82.04 |
| | Rounded Optimize | $83.03 \pm 1.1$ | $83.55 \pm 1.2$ | $82.51 \pm 1.3$ | 83.03 |
| | Normalized Optimize | $\mathbf{85.85} \pm 0.7$ | $\mathbf{88.56} \pm 0.8$ | $\mathbf{83.04} \pm 0.8$ | **85.79** |
| | $\mathcal{L}_{align}$ Only | $40.24 \pm 0.05$ | $47.22 \pm 0.1$ | $33.04 \pm 0.02$ | 40.13 |
| | $\mathcal{L}_{retain}$ Only | $84.48 \pm 1.2$ | $87.70 \pm 0.6$ | $81.16 \pm 0.5$ | 84.43 |

Table 2: Different optimization strategies and ablation study.

Considering the two optimization objectives we have designed: $\mathcal{L}_{align}$ and $\mathcal{L}_{retain}$, the actual loss values computed for these two are not on the same scale. Therefore, if gradient descent is directly applied, the overall optimization direction will be dominated by the objective with the larger loss value, leading to failure in achieving our desired effects. To address this, we adopt and compare several common balancing methods in multi-task learning. Specifically, **"Weighted Optimize"** refers to balancing the two losses through manually set weights, which are set as $\omega_a = 0.4$ and $\omega_r = 0.6$ in our experiments. **"Rounded Optimize"** means alternately optimizing one of the two losses in different epochs to mitigate the mutual influence during the optimization of the two losses; in this case, we alternate every epoch. As for **"Normalized Optimize"**, it indicates normalizing each loss value using its own value after each loss calculation, i.e., setting $\omega_a = \frac{1}{\|\mathcal{L}_{align}\|}$ and $\omega_r = \frac{1}{\|\mathcal{L}_{retain}\|}$, so that each loss is normalized to a unified scale for better convergence. Meanwhile, to demonstrate the necessity of combining and optimizing both align and retain losses, we also independently tested the two optimization objectives($\mathcal{L}_{align}$ Only and $\mathcal{L}_{retain}$ Only).

The representative experimental results can be derived from subsection 4.2(More detailed parametric experiments are provided in Appendix E), which indicates that the **Normalized Optimize** method

---

[1] https://github.com/samuela/git-re-basin
[2] https://github.com/gstoica27/ZipIt

achieved the best performance, whereas directly applying the weighted method or the rounded optimization approach failed to yield better outcomes. Additionally, optimizing using only one component of the objective function did not attain the optimal results achieved through joint optimization. This, to some extent, demonstrates that the mutual constraints (or adversarial) between the two distinct optimization objectives can facilitate learning more valuable permutations. It is noteworthy that using only $\mathcal{L}_{retain}$ yielded decent results, suggesting that valuable permutations can be learned directly from the data; however, the best performance was attained only through the constraints imposed by $\mathcal{L}_{align}$.

### 4.3 Fusion of Task Models with Different Distributions

| Fusion Method | Fused Model | MNIST | Fashion | KMNIST | Avg |
|---|---|---|---|---|---|
| Naive | MNIST | $95.58 \pm 0.01$ | $9.81 \pm 0.01$ | $9.70 \pm 0.01$ | 38.36 |
| | Fashion | $13.25 \pm 0.01$ | $96.78 \pm 0.01$ | $8.82 \pm 0.01$ | 39.61 |
| | KMNIST | $3.40 \pm 0.01$ | $18.84 \pm 0.01$ | $99.27 \pm 0.01$ | 40.50 |
| Weight Interpolation | MNIST + Fashion | $11.66 \pm 0.01$ | $59.43 \pm 0.01$ | $9.48 \pm 0.01$ | 26.85 |
| | MNIST+KMNIST | $9.65 \pm 0.01$ | $15.09 \pm 0.01$ | $60.27 \pm 0.01$ | 28.34 |
| | KMNIST+Fashion | $9.04 \pm 0.01$ | $12.09 \pm 0.01$ | $14.04 \pm 0.01$ | 11.72 |
| | Fused ALL | $9.40 \pm 0.01$ | $10.05 \pm 0.01$ | $9.66 \pm 0.01$ | 9.70 |
| Git Re-basin | MNIST + Fashion | $12.36 \pm 0.2$ | $10.32 \pm 1.7$ | $20.48 \pm 0.8$ | 14.29 |
| | MNIST+KMNIST | $10.23 \pm 0.3$ | $9.88 \pm 0.1$ | $15.58 \pm 0.1$ | 11.89 |
| | KMNIST+Fashion | $10.12 \pm 0.4$ | $12.92 \pm 0.1$ | $19.16 \pm 0.3$ | 14.06 |
| | Fused ALL | $10.29 \pm 0.6$ | $9.11 \pm 1.1$ | $13.76 \pm 0.9$ | 11.05 |
| Zipit | MNIST + Fashion | $10.75 \pm 1.1$ | $12.23 \pm 2.7$ | $21.92 \pm 2.8$ | 14.97 |
| | MNIST+KMNIST | $15.41 \pm 1.2$ | $9.11 \pm 0.1$ | $24.95 \pm 0.8$ | 16.49 |
| | KMNIST+Fashion | $10.42 \pm 1.1$ | $14.45 \pm 0.9$ | $23.79 \pm 1.9$ | 16.22 |
| | Fused ALL | $9.98 \pm 0.1$ | $9.12 \pm 0.1$ | $10.87 \pm 0.4$ | 9.99 |
| AutoFusion | MNIST + Fashion | $66.40 \pm 0.9$ | $\mathbf{86.20} \pm 0.8$ | $8.32 \pm 0.6$ | 53.64 |
| | MNIST+KMNIST | $72.19 \pm 1.1$ | $17.85 \pm 1.3$ | $\mathbf{93.44} \pm 2.1$ | 61.16 |
| | KMNIST+Fashion | $6.71 \pm 1.8$ | $80.58 \pm 1.0$ | $88.83 \pm 1.3$ | 58.70 |
| | Fused ALL | $\mathbf{86.99} \pm 2.4$ | $73.84 \pm 3.3$ | $67.09 \pm 2.8$ | $\mathbf{75.97}$ |

Table 3: Fusion of different distribution models.

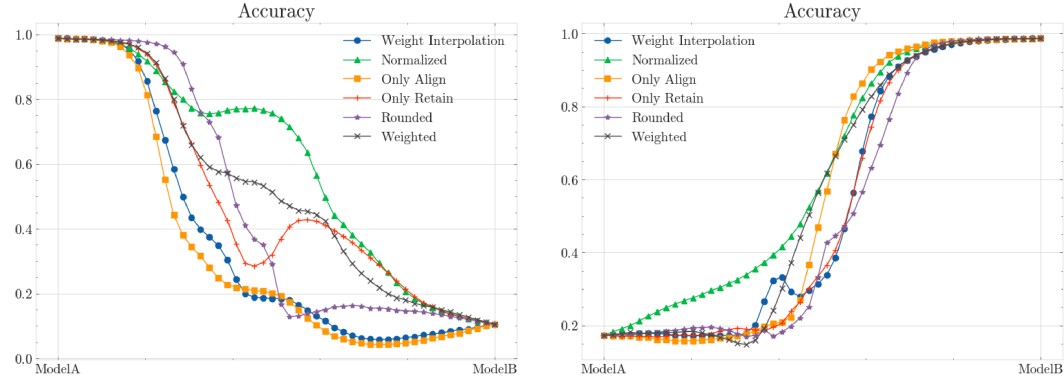

Figure 3: The interpolation test of each model on task A and task B after parameter fusion is carried out through the permutation matrices learned from different optimization objectives.

To further test the potential of AutoFusion, we proposed a more challenging experimental setup. In previous experiments, Task A and Task B were both from the same distribution. However, in this experiment, we chose datasets with completely different source distributions to test the ability of AutoFusion to fuse multi-task models trained on different distribution datasets. We selected MNIST LeCun et al. (1998), Fashion-MNIST (referred to as Fashion) Xiao et al. (2017), and KMNIST Clanuwat et al. (2018) datasets. After independently training models on these three datasets, we tested the performance of pairwise fusion models and the fusion of all three models together. The experimental results are shown in subsection 4.3 It is evident that after fusing the three models together, AutoFusion achieved an average accuracy of 75.97% across the three datasets, which is approximately 65% higher than the baseline Weight Interpolation method. Additionally, AutoFusion

achieved good results in pairwise model fusion. Particularly, after fusing the three models, the performance on the MNIST dataset was higher than any pairwise fusion models, indicating that the fusion process enabled the model to extract features better suited for the MNIST dataset. This once again demonstrates that the AutoFusion method learns meaningful parameter permutations. The specific experimental setup is provided in subsection D.4.

### 4.4 VISUALIZATION

In this section, we provide some visualizations of the results to facilitate a more comprehensive understanding of our method. Here we only show the visualization results of the linear interpolation experiment to fully demonstrate the stability and superiority of the AutoFusion method under all interpolation coefficients, more visualizations are given in Appendix F.

**Linear Interpolation**: Interpolating the parameters of two models to be fused using different interpolation parameters($\gamma \in [0, 1]$) and evaluating the interpolated fusion model on a test set can observe the loss barriers between the two models Ainsworth et al. (2022) Peña et al. (2023) Navon et al. (2023). In this work, we extend this visualization method to multi-task evaluation (subsection D.5 for detailed settings). For two models trained on different tasks, we set up three visualization perspectives. Two of them are the accuracies of the interpolated models, obtained through different interpolation parameters, on the test sets of Task A and Task B Figure 7, respectively. The third one is the test accuracy of the interpolated

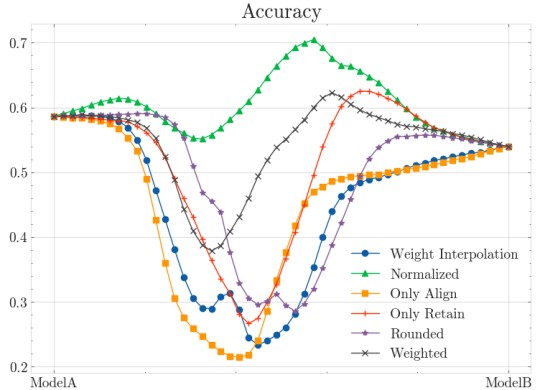

Figure 4: The interpolation test on joint dataset.

model on a complete dataset integrating both Task A and Task B Figure 4. It can be observed that when considering Task A and Task B separately, AutoFusion with Normalized has a higher accuracy rate than other settings. More strikingly, when tested on the integrated multi-task dataset, AutoFusion with Normalized shows a sharp contrast to the direct parameter interpolation method. When the interpolation parameter is around $0.6$, the accuracy of the direct interpolation method reaches its lowest point, while the accuracy of our method peaks, with a difference in accuracy exceeding $50\%$. This strongly demonstrates that our method can effectively fuse two models trained on different tasks using learnable permutations, enabling the fused model to exhibit promising performance in multi-task completion.

## 5 LIMITATION

Currently, most research on parameter fusion testing remains confined to simple models and datasets, and this paper is no exception. Existing methods have yet to yield significant results on complex datasets. Furthermore, because the automated fusion proposed by AutoFusion cannot be completely detached from data, we had to sample some training data to learn the permutation matrix. In the future, it may be possible to guide model fusion through fixed sets of data, but such endeavors will have to be left to future researchers.

## 6 CONCLUSION

This paper proposes a method named AutoFusion, which can learn a permutation matrix using only a few input samples. This permutation matrix effectively merges the parameters of two models designed for distinct tasks, resulting in a fused model capable of handling multiple tasks while maximizing accuracy across those tasks. AutoFusion can be regarded as the first work to apply an end-to-end approach in the field of multi-task parameter fusion. It significantly overcomes the bottleneck of previous works that heavily relied on prior knowledge and provides a valuable paradigm for the subsequent development of multi-task parameter fusion.

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

APPENDIX

## A    REPRODUCIBILITY STATEMENT

Our research work has been completed and the code has been organized. After the double-blind review process, we will open-source the code on GitHub to ensure the reproducibility of our research results. We welcome other researchers to replicate our methods and experiments.

## B    ETHICS STATEMENT

This study follows the ethical guidelines in the field of computer science. We use publicly available datasets in our research, all of which have been publicly released and comply with relevant usage agreements. We will respect the intellectual property rights of the original creators and contributors of the datasets and strictly adhere to data usage agreements and regulations. In the process of data processing and analysis, we will ensure proper handling of data accuracy and integrity, and avoid discrimination and bias. We commit to complying with relevant laws, regulations, and research ethics guidelines, and will not misuse data or disclose personal information.

## C    THEORETICAL ANALYSIS ABOUT SINKHORN

The use of the Sinkhorn operator in our AutoFusion method is grounded in its ability to approximate the discrete permutation problem in a continuous, differentiable manner. This section presents a theoretical analysis to justify its adoption and establish error bounds for the approximation.

### C.1    APPROXIMATION ERROR ANALYSIS

Consider the discrete permutation matrix $P^*$ that maximizes the inner product with the matrix $X$ subject to the entropy regularization. The Sinkhorn operator, $S_\tau(X)$, provides a continuous relaxation of this problem. We aim to bound the error between the discrete optimal permutation matrix $P^*$ and the soft permutation matrix $S_\tau(X)$ obtained from the Sinkhorn operator.

Let $\mathcal{E}$ denote the approximation error:

$$\mathcal{E} = \langle P^* - S_\tau(X), X \rangle_F \tag{23}$$

where $\langle \cdot, \cdot \rangle_F$ is the Frobenius inner product.

**Theorem (Error Bound for the Sinkhorn Operator)**: For any fixed $\tau > 0$, the approximation error $\mathcal{E}$ satisfies the following inequality:

$$\mathcal{E} \leq \frac{\|X\|_\infty^2}{2\tau} \tag{24}$$

*Proof:* We begin by observing that the Sinkhorn operator can be expressed as a fixed point iteration:

$$S_\tau(X) = \lim_{t \to \infty} S_\tau^{(t)}(X) \tag{25}$$

where $S_\tau^{(t)}(X)$ is the $t$-th iteration of the soft Sinkhorn operator as defined in Equation 13.

The entropy-regularized optimal transport problem can be rewritten as a fixed point problem:

$$P^* = \mathcal{T}_c(\mathcal{T}_r(P^* \exp(X/\tau))) \tag{26}$$

By the triangle inequality, we have:

$$\mathcal{E} \leq \langle P^* - S_\tau^{(t)}(X), X \rangle_F + \langle S_\tau^{(t)}(X) - S_\tau^{(t+1)}(X), X \rangle_F \tag{27}$$

The second term can be bounded using the update rule of the Sinkhorn operator:

$$\langle S_\tau^{(t)}(X) - S_\tau^{(t+1)}(X), X \rangle_F \leq \frac{\|X\|_\infty^2}{2\tau} \tag{28}$$

Taking the limit as $t \to \infty$, we obtain the desired error bound.

This theorem establishes that the approximation error is upper-bounded by a quantity that depends on the matrix norm $\|X\|_\infty$ and the regularization parameter $\tau$. As $\tau$ increases, the error bound becomes tighter, indicating that the soft permutation matrix approaches the optimal discrete permutation matrix.

## C.2 Differentiability and Gradient Flow

**Proposition (Differentiability of the Sinkhorn Operator)**: The Sinkhorn operator $S_\tau^{(t)}(X)$ is differentiable with respect to $X$ for all $t \geq 0$, and its derivative can be expressed as:

$$\frac{\partial S_\tau^{(t)}(X)}{\partial X} = \frac{\partial S_\tau^{(t)}(X)}{\partial P^{(t)}} \frac{\partial P^{(t)}}{\partial X} \tag{29}$$

where $\frac{\partial S_\tau^{(t)}(X)}{\partial P^{(t)}}$ is the Jacobian of the Sinkhorn operator with respect to the intermediate iterate $P^{(t)}$, and $\frac{\partial P^{(t)}}{\partial X}$ is the derivative of the intermediate iterate with respect to $X$.

The proof follows from the chain rule and the differentiability of the row and column normalization operations.

## D Implementation Details

### D.1 Description of Datasets Used Above

**MNIST** The MNIST dataset comes from the National Institute of Standards and Technology (NIST) in the United States. The training set consists of handwritten digits from 250 different individuals, with 50% from high school students and 50% from employees of the Census Bureau. The test set also contains handwritten digits in the same proportions, but authors of the test set do not overlap with those of the training set. The MNIST dataset comprises a total of 70,000 images, with 60,000 images in the training set and 10,000 images in the test set. Each image is a 28x28 pixel grayscale image representing a handwritten digit from 0 to 9. The images have a black background represented by 0 and the white digits are represented by floating-point values between 0 and 1, where values closer to 1 indicate a whiter color.

**CIFAR-10** The CIFAR-10 dataset consists of 60,000 samples, each of which is a 32x32 pixel RGB image (color image). Each RGB image is divided into three channels (R channel, G channel, B channel). These 60,000 samples are divided into 50,000 training samples and 10,000 test samples. CIFAR-10 contains 10 classes of objects, labeled from 0 to 9, representing airplane, automobile, bird, cat, deer, dog, frog, horse, ship, and truck.

**Fashion-MNIST** Fashion-MNIST is an image dataset that serves as a replacement for the MNIST handwritten digit dataset. It was provided by the research department of Zalando, a fashion technology company based in Germany. The dataset consists of 70,000 frontal images of 10 different categories of items. The size, format, and division of training and test sets in Fashion-MNIST are identical to the original MNIST dataset. The dataset is divided into 60,000 training samples and 10,000 test samples, with 28x28 grayscale images that can be directly used for training models designed for MNIST.

**KMNIST** KMNIST is derived from Japanese Hiragana and Katakana characters and is maintained and open-sourced by the ROIS-DS Center for Open Data in the Humanities. The dataset consists of 70,000 high-resolution handwritten samples, with 10,000 samples per class, totaling 46 different character types. The purpose of KMNIST is to serve as a Japanese version of the MNIST dataset, used to evaluate the capabilities of machine learning and deep learning models in multi-language text recognition tasks.

For all datasets, we extracted 1000 images as a validation set for parameter tuning. These images were divided from the original training set and do not affect the test set.

### D.2 Details for Comparison Experiments

When independently training models on the divided datasets, we used the classic CNN and MLP architecture, for CNN, we used VGG to extract features and for MLP we designed 6 hidden layers to extract features. For simpler datasets like MNIST, we chose a smaller VGG model, while for more complex datasets like CIFAR-10, we used a deeper VGG to extract higher-level features. During training, the learning rate was set to $0.01$ for the MNIST dataset and $0.001$ for the CIFAR-10 dataset, with cross-entropy loss used as the training objective. When applying the method in subsection 3.1

to perform a differentiable approximation of the Sinkhorn operator, we found that using a limited number of iterations could achieve good approximation results. Therefore, we set the iteration coefficient $t = 20$ when training the permutation matrix. For the data needed to train the matrix, we randomly sampled 2000 input examples from the MNIST dataset and 2000 input examples from the CIFAR-10 dataset. When selecting pseudo-labels, we set the filtering threshold $\zeta = 0.9$ to filter out samples with lower confidence. Regarding the choice of combination weights for $\mathcal{L}_{align}$ and $\mathcal{L}_{retain}$ losses, due to the different magnitudes of the two losses, their impacts on parameter gradients were different. We ultimately adopted the commonly used technique in multi-task training to normalize the losses, scaling them to around 1, to achieve more stable optimization results.

### D.3 DETAILS FOR ABLATION STUDY

We conducted evaluations on two architectures, CNN(VGG) and MLP, using the MNIST dataset. For the **"Weighted Optimize"** test, to avoid over-tuning and overfitting the model to the test set, we empirically chose the values $\omega_a = 0.5$ and $\omega_r = 0.5$. We then learned the parameter permutation matrix under these parameters, performed parameter fusion, and evaluated the final results. For the **"Rounded Optimize"** test, we employed a method of switching optimization objectives at each epoch. During the optimization matrix learning process, the learning rate was initially set to 1 and then decreased to 0.01 using a cosine annealing scheduler after 64 rounds to achieve convergence.

### D.4 DETAILS FOR MULTI-TASK FUSION

In conducting non-identically distributed multi-task fusion, we conducted two experiments. The first experiment involved fusing models trained on two datasets. Apart from differences in data sources and the number of classes, the fusion steps in this experiment were the same as those in the comparison experiment, with no further elaboration. The second experiment involved fusing models trained on three datasets. Due to the limitations of the AutoFusion method, which can only learn one permutation matrix and fuse one model at a time, we first fused two models, saved the fused parameters, and then learned the permutation of the third model to better fuse with the saved parameters. This resulted in a fused model of three models. In this experiment, we prioritized fusing the models trained on MNIST and Fashion-MNIST, then learned the permutation of KMNIST to fuse it with the parameters obtained from the fusion of the first two models. Training the final permutation required accessing partial training sets from all three models. In this study, we sampled 10% of each training set to ensure the effectiveness of multi-task fusion. By following this approach, the AutoFusion method can actually fuse more model parameters.

### D.5 DETAILS FOR LINEAR INTERPOLATION

To better demonstrate the differences between different optimization strategies in linear interpolation, as well as the overall effectiveness of the AutoFusion method in the entire interpolation space, we selected the relatively simple MNIST dataset and the CNN (VGG) architecture. When training the permutation matrix, we used randomly sampled 1000 samples along with their true labels, with a sampling seed set at 3315. After learning the permutation matrix using different methods, we uniformly sampled 50 points in the range $[0, 1]$. These points were used as values for $\gamma$ in turn, and models were fused using linear interpolation method. The final results were obtained on various test sets.

## E MORE EXPERIMENTS

### E.1 PARAMETRIC EXPERIMENTS ON PSEUDO-LABEL SELECTION THRESHOLD

To thoroughly validate the impact of pseudo-label threshold selection on the final results when conducting unsupervised permutation matrix learning, we conducted a fusion experiment using the AutoFusion method on the CNN (VGG) model on the MNIST dataset. Apart from the pseudo-label threshold, all other parameter settings were consistent with those of subsection D.2. The results are shown in subsection E.1

We still divide the 10-class MNIST dataset into two five-class datasets, represented as Dataset A and Dataset B respectively. In the experimental results, Task A represents the accuracy tested on Dataset A, while Task B represents the accuracy tested on Dataset B. Joint indicates the evaluation results on the complete dataset, while Avg represents the simple arithmetic average of Task A and Task B. The difference between **"with Weighted"** and **"without Weighted"** in the table lies in whether layer-wise weighting is applied when calculating $\mathcal{L}_{align}$. It can be observed that as $\zeta$ increases, the overall trend of the AutoFusion method is a gradual increase in the accuracy of Joint, reflecting the significant impact of the pseudo-label threshold on the results, and indicating that this method can effectively filter out incorrect labels. By comparing the results of the AutoFusion method with and without layer-wise weighting, it can be seen that layer-wise weighted averaging performs better. The reason for this may be that the neurons in the shallow layers of neural networks generally learn more universal features, making them more suitable for alignment. Therefore, assigning greater weight to align the neurons in the shallow layers during weighting contributes to the learning of higher-quality alignment matrices.

| Method | $\zeta$ | Joint | TaskA | TaskB | Avg |
|---|---|---|---|---|---|
| Model A | - | $58.65 \pm 0.01$ | $98.77 \pm 0.01$ | $17.31 \pm 0.01$ | 58.03 |
| Model B | - | $53.97 \pm 0.01$ | $10.60 \pm 0.01$ | $98.63 \pm 0.01$ | 54.61 |
| Weight Interpolation | - | $25.44 \pm 0.01$ | $18.58 \pm 0.01$ | $32.50 \pm 0.01$ | 25.54 |
| **AutoFusion** with Weighted | 0.9 | $65.53 \pm 0.2$ | $58.47 \pm 0.1$ | $72.79 \pm 0.4$ | 65.63 |
| | 0.8 | $64.61 \pm 0.3$ | $57.39 \pm 0.4$ | $72.04 \pm 0.5$ | 64.72 |
| | 0.7 | $65.88 \pm 1.1$ | $60.13 \pm 1.1$ | $71.80 \pm 0.6$ | 65.96 |
| | 0.6 | $64.18 \pm 0.6$ | $56.58 \pm 0.7$ | $72.00 \pm 0.6$ | 64.29 |
| | 0.5 | $63.59 \pm 0.5$ | $57.58 \pm 0.5$ | $69.77 \pm 0.4$ | 63.68 |
| | 0.4 | $63.26 \pm 0.6$ | $55.97 \pm 0.5$ | $70.77 \pm 0.8$ | 63.37 |
| | 0.3 | $64.72 \pm 0.4$ | $59.18 \pm 0.3$ | $70.42 \pm 0.5$ | 64.80 |
| | 0.2 | $64.42 \pm 0.7$ | $57.04 \pm 0.9$ | $72.03 \pm 0.8$ | 64.54 |
| | 0.1 | $63.72 \pm 1.1$ | $55.40 \pm 1.2$ | $72.28 \pm 0.7$ | 63.84 |
| **AutoFusion** without Weighted | 0.9 | $65.28 \pm 0.3$ | $59.91 \pm 0.4$ | $70.81 \pm 0.5$ | 65.36 |
| | 0.8 | $61.66 \pm 0.3$ | $53.07 \pm 0.3$ | $70.50 \pm 0.3$ | 65.36 |
| | 0.7 | $60.54 \pm 0.4$ | $49.78 \pm 0.7$ | $71.62 \pm 0.5$ | 60.70 |
| | 0.6 | $62.55 \pm 0.5$ | $57.56 \pm 0.7$ | $67.68 \pm 0.3$ | 62.62 |
| | 0.5 | $61.43 \pm 0.4$ | $53.98 \pm 0.4$ | $69.10 \pm 0.5$ | 61.54 |
| | 0.4 | $63.86 \pm 1.2$ | $55.47 \pm 0.9$ | $72.49 \pm 1.9$ | 63.98 |
| | 0.3 | $60.42 \pm 0.6$ | $53.68 \pm 0.7$ | $67.35 \pm 1.2$ | 60.52 |
| | 0.2 | $61.87 \pm 0.8$ | $52.40 \pm 0.6$ | $71.61 \pm 1.3$ | 62.01 |
| | 0.1 | $59.62 \pm 0.6$ | $52.96 \pm 0.3$ | $66.48 \pm 0.9$ | 59.72 |

Table 4: The impact of different pseudo-label thresholds on the final result.

### E.2 PARAMETRIC EXPERIMENTS ON WEIGHTED OPTIMIZE

In this experiment, we primarily investigated the optimization of alignment matrices by directly weighting two loss functions ($\mathcal{L}_{align}$ and $\mathcal{L}_{retain}$). We continued to utilize the CNN (VGG) architecture model and the MNIST dataset, with the same data partitioning method and parameter settings as subsection D.2. In this study, our focus was solely on analyzing the performance of the fused model when different weights were applied to the loss functions. As can be seen from subsection E.2, it is evident that with varying weighting methods, the values of Joint only fluctuated within a certain range, indicating the model's stability with respect to this parameter. All results outperformed Weight Interpolation by approximately 40% in terms of accuracy, further emphasizing the stability of the AutoFusion method.

### E.3 PARAMETRIC EXPERIMENTS ON ROUNDED OPTIMIZE

In this experiment, we investigated the impact of optimizing different loss functions on the AutoFusion method based on epochs. Specifically, we optimized either $\mathcal{L}_{align}$ or $\mathcal{L}_{retain}$ during a single backpropagation, switching the optimization target every n epochs. The experiment utilized the

| Method | Weights | Joint | TaskA | TaskB | Avg |
|--------|---------|-------|-------|-------|-----|
| Model A | - | $58.65 \pm 0.01$ | $98.77 \pm 0.01$ | $17.31 \pm 0.01$ | 58.03 |
| Model B | - | $53.97 \pm 0.01$ | $10.60 \pm 0.01$ | $98.63 \pm 0.01$ | 54.61 |
| Weight Interpolation | - | $25.44 \pm 0.01$ | $18.58 \pm 0.01$ | $32.50 \pm 0.01$ | 25.54 |
| **AutoFusion** | $\omega_a = 0.1, \omega_r = 0.9$ | $64.09 \pm 1.6$ | $57.68 \pm 0.8$ | $70.68 \pm 1.2$ | 64.18 |
| | $\omega_a = 0.2, \omega_r = 0.8$ | $63.91 \pm 0.8$ | $54.33 \pm 0.9$ | $73.77 \pm 1.2$ | 64.05 |
| | $\omega_a = 0.3, \omega_r = 0.7$ | $63.94 \pm 0.6$ | $55.16 \pm 0.5$ | $72.98 \pm 0.9$ | 64.07 |
| | $\omega_a = 0.4, \omega_r = 0.6$ | $61.27 \pm 1.2$ | $52.93 \pm 0.8$ | $69.85 \pm 0.7$ | 61.39 |
| | $\omega_a = 0.5, \omega_r = 0.5$ | $61.12 \pm 1.1$ | $51.51 \pm 0.8$ | $71.01 \pm 0.9$ | 61.26 |
| | $\omega_a = 0.6, \omega_r = 0.4$ | $60.12 \pm 0.6$ | $50.82 \pm 0.5$ | $69.69 \pm 0.6$ | 60.26 |
| | $\omega_a = 0.7, \omega_r = 0.3$ | $62.79 \pm 0.4$ | $53.94 \pm 0.8$ | $71.90 \pm 1.2$ | 62.92 |
| | $\omega_a = 0.8, \omega_r = 0.2$ | $64.49 \pm 1.3$ | $57.67 \pm 0.8$ | $71.51 \pm 1.5$ | 64.59 |
| | $\omega_a = 0.9, \omega_r = 0.1$ | $63.21 \pm 0.3$ | $56.52 \pm 0.4$ | $70.09 \pm 0.5$ | 63.31 |

Table 5: Different Weight Setting for Weighted Optimize Strategies.

CNN (VGG) network on the MNIST dataset, with the same data partitioning method and parameter settings as subsection D.2. We tested the performance of AutoFusion with different switching cycles (i.e., switching optimization targets at different epoch intervals). The experimental results, as shown in subsection E.3, revealed that the optimal result of 65.19% accuracy was achieved when the epoch was set to 5. However, the performance of AutoFusion did not vary significantly with changes in the epoch interval. Nonetheless, it consistently outperformed the Weight Interpolation method by a significant margin, demonstrating the stability of AutoFusion in response to parameter variations.

| Method | Epochs | Joint | TaskA | TaskB | Avg |
|--------|--------|-------|-------|-------|-----|
| Model A | - | $58.65 \pm 0.01$ | $98.77 \pm 0.01$ | $17.31 \pm 0.01$ | 58.03 |
| Model B | - | $53.97 \pm 0.01$ | $10.60 \pm 0.01$ | $98.63 \pm 0.01$ | 54.61 |
| Weight Avg | - | $25.44 \pm 0.01$ | $18.58 \pm 0.01$ | $32.50 \pm 0.01$ | 25.54 |
| **AutoFusion** | $epoch = 1$ | $62.36 \pm 1.2$ | $53.88 \pm 1.1$ | $71.09 \pm 1.3$ | 62.48 |
| | $epoch = 2$ | $62.11 \pm 0.6$ | $54.55 \pm 0.7$ | $69.89 \pm 1.1$ | 62.22 |
| | $epoch = 3$ | $63.08 \pm 0.8$ | $55.32 \pm 0.4$ | $71.07 \pm 0.7$ | 63.19 |
| | $epoch = 4$ | $64.78 \pm 0.4$ | $55.38 \pm 0.2$ | $74.46 \pm 0.6$ | 64.92 |
| | $epoch = 5$ | $65.19 \pm 0.4$ | $58.98 \pm 0.3$ | $71.57 \pm 0.3$ | 65.27 |
| | $epoch = 6$ | $62.12 \pm 1.2$ | $52.71 \pm 0.7$ | $71.80 \pm 1.1$ | 62.26 |
| | $epoch = 7$ | $61.99 \pm 1.1$ | $54.78 \pm 0.3$ | $69.40 \pm 0.9$ | 62.09 |
| | $epoch = 8$ | $61.07 \pm 1.7$ | $49.70 \pm 2.2$ | $72.77 \pm 1.2$ | 61.24 |

Table 6: Different Epoch Setting for Rounded Optimize Strategies.

### E.4 NUMBER OF STEP TO APPROXIMATE THE SINKHORN OPERATOR

Although we provided an upper bound on the error of approximating the sinkhorn operator using an iterative approach in the paper, it is still essential to explore the impact of the approximation steps on the results in experiments. In this experiment, we set different numbers of iteration steps and recorded the performance of the AutoFusion method at the corresponding iteration steps. We continued to use the CNN (VGG) network as the feature extraction network and selected the MNIST dataset, keeping the settings of other parameters and data partitioning method consistent with subsection D.2, with the only variation being the number of iteration steps. The experimental results, as shown in subsection E.4, indicate that when the number of iterations is generally large, the accuracy of AutoFusion on Joint is relatively high. However, when the number of iterations exceeds 20, the accuracy of Joint does not show significant improvement with an increase in the number of iterations. In fact, the accuracy of Joint may even decrease due to the impact of the iteration steps on convergence speed.

| Method | Iteration | Joint | TaskA | TaskB | Avg |
|---|---|---|---|---|---|
| Model A | - | $58.65 \pm 0.01$ | $98.77 \pm 0.01$ | $17.31 \pm 0.01$ | 58.03 |
| Model B | - | $53.97 \pm 0.01$ | $10.60 \pm 0.01$ | $98.63 \pm 0.01$ | 54.61 |
| Weight Avg | - | $25.44 \pm 0.01$ | $18.58 \pm 0.01$ | $32.50 \pm 0.01$ | 25.54 |
| **AutoFusion** | $iter = 5$ | $61.29 \pm 1.4$ | $52.13 \pm 1.3$ | $70.72 \pm 1.1$ | 61.43 |
| | $iter = 10$ | $63.41 \pm 1.0$ | $56.97 \pm 0.9$ | $70.03 \pm 1.2$ | 63.50 |
| | $iter = 15$ | $64.04 \pm 0.5$ | $57.23 \pm 0.3$ | $71.05 \pm 1.1$ | 64.14 |
| | $iter = 20$ | $65.62 \pm 0.6$ | $58.47 \pm 0.2$ | $72.79 \pm 1.3$ | 65.63 |
| | $iter = 25$ | $64.50 \pm 0.4$ | $57.92 \pm 0.3$ | $71.27 \pm 0.9$ | 64.59 |
| | $iter = 30$ | $64.98 \pm 0.3$ | $56.36 \pm 0.2$ | $73.85 \pm 1.2$ | 65.10 |
| | $iter = 35$ | $63.77 \pm 0.5$ | $55.06 \pm 0.5$ | $72.73 \pm 0.8$ | 63.99 |
| | $iter = 40$ | $62.63 \pm 0.4$ | $54.69 \pm 0.3$ | $70.80 \pm 1.2$ | 62.75 |

Table 7: Different Step to Approximate the Sinkhorn Operator.

### E.5 TRAINING PERMUTATION MATRICES AT DIFFERENT DATA USAGE RATIOS USING REAL LABELS

In order to fully explore the potential of the AutoFusion method, we attempted to train parameter alignment matrices using real data labels and tested the relationship between the size of the training data and the performance of the fusion model. In this experiment, we utilized the CNN (VGG) architecture on the MNIST dataset. We initially extracted 9000 images from MNIST as the complete dataset, where the "Part" column in the table represents the proportion of the complete dataset used. All data had access to real labels. The experimental results in subsection E.5 clearly demonstrate that compared to previous experiments without access to real data labels, training with real labels resulted in better alignment matrix learning. With only 10% of real data, the Joint accuracy of the fusion model reached 73.06%. As the proportion of data increased, the Joint accuracy peaked at 83.21%. This indicates that the hypothesis proposed by the AutoFusion method, which suggests that **learning parameter alignments can facilitate multi-task parameter fusion, is correct**. It also suggests that there exists an opportunity in the future to approximate the optimal alignment matrix through more advanced algorithm designs. An interesting observation in the experimental results is that when the proportion of data used exceeded 40%, there was not a significant increase in Joint accuracy. This further highlights that **AutoFusion operates within a limited search space and can yield valuable solutions even when using a small portion of the data**.

| Method | Part | Joint | TaskA | TaskB | Avg |
|---|---|---|---|---|---|
| Model A | - | $58.65 \pm 0.01$ | $98.77 \pm 0.01$ | $17.31 \pm 0.01$ | 58.03 |
| Model B | - | $53.97 \pm 0.01$ | $10.60 \pm 0.01$ | $98.63 \pm 0.01$ | 54.61 |
| Weight Interpolation | - | $25.44 \pm 0.01$ | $18.58 \pm 0.01$ | $32.50 \pm 0.01$ | 25.54 |
| **AutoFusion** | 10% | $73.06 \pm 1.1$ | $66.81 \pm 0.9$ | $79.49 \pm 1.2$ | 73.15 |
| | 20% | $77.39 \pm 0.4$ | $71.95 \pm 0.3$ | $82.98 \pm 0.6$ | 77.47 |
| | 30% | $77.79 \pm 0.3$ | $73.17 \pm 0.3$ | $82.54 \pm 0.4$ | 77.86 |
| | 40% | $82.55 \pm 0.7$ | $75.40 \pm 0.6$ | $89.91 \pm 0.9$ | 82.66 |
| | 50% | $79.29 \pm 0.5$ | $74.77 \pm 0.2$ | $83.94 \pm 0.7$ | 79.36 |
| | 60% | $80.23 \pm 1.2$ | $73.52 \pm 0.3$ | $87.21 \pm 0.8$ | 80.37 |
| | 70% | $81.19 \pm 0.3$ | $73.66 \pm 0.4$ | $89.70 \pm 0.3$ | 81.68 |
| | 80% | $82.88 \pm 1.2$ | $78.12 \pm 0.9$ | $87.77 \pm 0.9$ | 82.95 |
| | 90% | $81.33 \pm 0.9$ | $76.19 \pm 0.3$ | $86.62 \pm 0.8$ | 81.41 |
| | 100% | $83.21 \pm 0.4$ | $78.24 \pm 0.3$ | $88.10 \pm 0.5$ | 83.17 |

Table 8: The impact of using different proportions of data on the effect of fusion.

## E.6 An exploration of the relationship between model depth and fusion effects

We also explored how the fusion model's capabilities using the AutoFusion method changed as the depth of the model gradually increased. In the following experiment subsection E.6, we selected the relatively simple but significantly effective MLP architecture as the feature extraction network, with the MNIST dataset still being utilized. The only variable in this experiment was the number of hidden layers in the MLP. Each hidden layer was a non-linear mapping layer ranging from 512 to 512. We set the number of these hidden layers to be 2, 4, 6, and 8. The initial model training process, data partitioning, and settings are consisted of subsection D.2. AutoFusion was used for parameter fusion at different depths as mentioned above. The experimental results indicate that as the number of hidden layers increased from 2 to 6, the accuracy of the Joint continued to improve. Since the original models of different depths had similar test results on the test set (ranging from 98% to 99%), the improvement in Joint accuracy after parameter fusion sufficiently demonstrates that **AutoFusion has better fusion capabilities for deeper networks**. However, when the number of hidden layers reached 8, we observed a slight decrease in Joint accuracy. This is likely due to the overfitting of the overly deep network to the MNIST dataset, which falls within the normal range of results.

| Method | Hidden Length | Joint | TaskA | TaskB | Avg |
|---|---|---|---|---|---|
| Model A | - | $58.65 \pm 0.01$ | $98.77 \pm 0.01$ | $17.31 \pm 0.01$ | 58.03 |
| Model B | - | $53.97 \pm 0.01$ | $10.60 \pm 0.01$ | $98.63 \pm 0.01$ | 54.61 |
| Weight Interpolation | - | $25.44 \pm 0.01$ | $18.58 \pm 0.01$ | $32.50 \pm 0.01$ | 25.54 |
| **AutoFusion** | $length = 2$ | $82.21 \pm 0.4$ | $89.59 \pm 0.8$ | $74.60 \pm 0.4$ | 82.09 |
| | $length = 4$ | $83.53 \pm 0.9$ | $88.45 \pm 1.1$ | $78.46 \pm 0.6$ | 83.46 |
| | $length = 6$ | $85.12 \pm 1.1$ | $82.99 \pm 0.9$ | $87.31 \pm 1.0$ | 85.15 |
| | $length = 8$ | $79.23 \pm 2.2$ | $70.89 \pm 1.7$ | $87.81 \pm 3.6$ | 79.35 |

Table 9: The impact of using different proportions of data on the effect of fusion.

## E.7 Evaluation of More Complex Models and Datasets

To further evaluate the generalization performance of the AutoFusion method on complex datasets as well as more complex models, we introduced the CIFAR100 dataset as well as the Resnet family of models. The results of the experiment are in Table 10, where CIFAR100-GS denotes the grayscale version of the CIFAR100 dataset. Our experimental setup remains consistent with previous experimental settings subsection D.2, and from these results it is clear that AutoFusion still maintains good generalization over more complex datasets as well as models.

## E.8 Testing AutoFusion on Object Detection Tasks

To further evaluate the generalization of the AutoFusion method, we tested the model fusion algorithm for the first time on a object detection task. Specifically, we utilize the Faster-RCNN method for object detection model training on the VOC2007 dataset, and we view the object detection model as a Feature layer as well as a Head layer, for the Feature layer, we use the pre-trained VGG16 network for feature extraction, which is a part of the parameters that we keep frozen throughout the training/fusion process, while the Head denotes the randomly initialized object detection head, which is also our target layer for training/fusion.

| Method | mAP |
|---|---|
| Model A | 24.64 |
| Model B | 25.43 |
| Ensemble | 55.24 |
| Git Re-basin | 20.99 |
| Zipit | 18.74 |
| AutoFusion | **36.02** |

Table 11: Testing model fusion on a VOC2007 object detection task

Since VOC2007 has 20 target detection classes, similar to the setup of the classification method, we divide these 20 classes into two parts, one containing 10 classes, and use the data from these two parts to train two models **Model A** as well as **Model B** meanwhile the **Ensemble** model is trained on the complete 20-class object detection training set to serve as an upper

| Setting | Method | Joint | TaskA | TaskB |
|---|---|---|---|---|
| CNN + CIFAR100-GS | Avg | 2.2 | 2.26 | 2.14 |
| | ModelA | 23.12 | 43.52 | 1.74 |
| | ModelB | 22.63 | 2.51 | 43.74 |
| | Git-Rebasin | 3.67 | 5.12 | 2.23 |
| | Zipit | 7.63 | 10.12 | 5.14 |
| | AutoFusion | **20.65** | **17.8** | **23.58** |
| CNN + CIFAR100 | Avg | 2.29 | 2.16 | 2.42 |
| | ModelA | 28.475 | 54.11 | 2.84 |
| | ModelB | 27.78 | 2.58 | 52.98 |
| | Git-Rebasin | 2 | 2.21 | 1.79 |
| | Zipit | 4.05 | 5.74 | 2.36 |
| | AutoFusion | **21.67** | **21.14** | **22.2** |
| Resnet18 + CIFAR100 | Avg | 2.28 | 2.45 | 2.1 |
| | ModelA | 27.03 | 51.06 | 3.11 |
| | ModelB | 30.13 | 2.88 | 57.38 |
| | Git-Rebasin | 1.69 | 2.27 | 1.11 |
| | Zipit | 4.51 | 6.79 | 2.22 |
| | AutoFusion | **32.85** | **35.62** | **30.08** |

Table 10: Comparative experiments between AutoFusion and baselines on complex datasets

bound reference value. We use different fusion methods to fuse the Head portion of Model A,B and perform them on the full test set Testing.

The overall experimental results are shown in Table 11. Thanks to the learning ability of the AutoFusion method, the mAP of the fusion model obtained when using the AutoFusion method for fusion is clearly higher than that of the other baseline models in the complete dataset, which further demonstrates the better adaptability of our method on different tasks.

In order to observe the effect of the AutoFusion fusion model in more detail, we display the detection information of the Ensemble model, Model A, Model B, and the AutoFusion fusion model on each target category in Table 12, Table 13. It can be clearly seen that the object detection models Model A and Model B, which were trained on some of the categories, have almost no detection effect on their own untrained categories, whereas the AutoFusion fused model, which incorporates the detection capabilities of each of the two models maintains a certain level of detection effect on all the categories, and at the same time maintains a certain level of detection effect in all categories, as compared to the Model A as well as Model B's The mAP metrics are very much improved, which further proves the effectiveness and scalability of the AutoFusion method.

## E.9 COMPUTATIONAL EFFICIENCY ANALYSIS

In order to further highlight the value of AutoFusion for generalized applications, we conducted an in-depth analysis of the computational efficiency of the algorithm. This analysis focuses on two main aspects, firstly on the trainable parameters, we compare the number of trainable parameters of AutoFusion in fusing two sub-models with the number of trainable parameters in directly training a new integrated model, but since AutoFusion actually learns a permutation matrix, the search space is much smaller than the general parameter training, so We also analyzed the convergence of the model, which is shown in Table 14.

We considered the model to have converged when the accuracy of the validation set on five adjacent training steps did not differ by more than 5% of its mean value. Based on this definition, we evaluated the difference in convergence speed between the ensemble model and AutoFusion under a batch size of 64 for each training step, and the **"Convergence Step"** in the table indicates the number of training steps at which the model converged. It is clear that AutoFusion guarantees very fast convergence at different settings, with only about 0.5% of the number of convergence steps of

| Ensemble | | | | | Model A | | | | |
|---|---|---|---|---|---|---|---|---|---|
| Class Name | AP | Recall | Precision | F1 | mAP | Class Name | AP | Recall | Precision | F1 | mAP |
| aeroplane | 49.91 | 75.93 | 24.4 | 0.37 | | aeroplane | 39.33 | 61.11 | 29.73 | 0.4 | |
| bicycle | 66.33 | 82.76 | 19.75 | 0.32 | | bicycle | 44.71 | 60.34 | 22.29 | 0.33 | |
| bird | 46.89 | 56.94 | 28.67 | 0.38 | | bird | 35.93 | 63.89 | 9.27 | 0.15 | |
| boat | 38.3 | 59.09 | 28.67 | 0.37 | | boat | 24.69 | 45.45 | 18.52 | 0.26 | |
| bottle | 48.54 | 70.83 | 26.53 | 0.36 | | bottle | 24.44 | 47.22 | 23.29 | 0.31 | |
| bus | 49.74 | 58.82 | 24.4 | 0.5 | | bus | 32.36 | 45.1 | 32.86 | 0.38 | |
| car | 73.23 | 85.95 | 43.48 | 0.47 | | car | 54.05 | 66.94 | 37.59 | 0.48 | |
| cat | 64.72 | 76.71 | 32.2 | 0.55 | | cat | 51.38 | 83.56 | 14.49 | 0.25 | |
| chair | 50.08 | 66.67 | 43.08 | 0.37 | | chair | 24.3 | 49.31 | 16.47 | 0.25 | |
| chow | 49.41 | 69.7 | 25.81 | 0.37 | 55.24 | chow | 34.64 | 51.52 | 20.24 | 0.29 | 24.64 |
| diningtable | 34.77 | 62.16 | 25.56 | 0.5 | | diningtable | 3.86 | 2.7 | 100 | 0.05 | |
| dog | 63.79 | 76.83 | 22.33 | 0.56 | | dog | 6.96 | 0 | 0 | 0.03 | |
| horse | 84.09 | 90.77 | 44.37 | 0.51 | | horse | 24.7 | 1.54 | 100 | 0.03 | |
| motorbike | 60.64 | 67.57 | 35.33 | 0.51 | | motorbike | 27.05 | 1.35 | 100 | 0.38 | |
| person | 70.99 | 84.61 | 40.98 | 0.48 | | person | 35.69 | 56.82 | 28.08 | 0 | |
| pottedplant | 34.36 | 49.57 | 33.1 | 0.37 | | pottedplant | 0.43 | 0 | 0 | 0 | |
| sheep | 66.28 | 72.55 | 29.44 | 0.53 | | sheep | 0 | 0 | 0 | 0 | |
| sofa | 38.63 | 68.97 | 42.05 | 0.28 | | sofa | 3.67 | 0 | 0 | 0 | |
| train | 49.4 | 73.68 | 17.86 | 0.36 | | train | 2.63 | 2.63 | 100 | 0.05 | |
| tvmonitor | 64.82 | 82.61 | 32.76 | 0.47 | | tvmonitor | 21.98 | 26.09 | 42.86 | 0.32 | |

Table 12: Fine-grained analysis of fusion object detection tasks using AutoFusion

| Model B | | | | | AutoFusion | | | | |
|---|---|---|---|---|---|---|---|---|---|
| Class Name | AP | Recall | Precision | F1 | mAP | Class Name | AP | Recall | Precision | F1 | mAP |
| aeroplane | 5 | 0 | 0 | 0 | | aeroplane | 38.55 | 48.15 | 29.89 | 0.37 | |
| bicycle | 27.11 | 53.45 | 21.83 | 0.31 | | bicycle | 46.34 | 67.24 | 29.32 | 0.41 | |
| bird | 0 | 0 | 0 | 0 | | bird | 38.88 | 62.5 | 20.74 | 0.31 | |
| boat | 1.79 | 0 | 0 | 0 | | boat | 18.07 | 15.91 | 36.84 | 0.22 | |
| bottle | 15.66 | 23.61 | 24.64 | 0.24 | | bottle | 28.58 | 23.61 | 54.84 | 0.33 | |
| bus | 28.05 | 47.06 | 27.59 | 0.35 | | bus | 34.68 | 39.22 | 40.82 | 0.4 | |
| car | 43.85 | 61.57 | 31.57 | 0.42 | | car | 49.29 | 72.31 | 21.63 | 0.33 | |
| cat | 6.34 | 0 | 0 | 0 | | cat | 56.61 | 75.34 | 35.26 | 0.48 | |
| chair | 14.96 | 26.39 | 25 | 0.26 | | chair | 19.03 | 9.72 | 40 | 0.16 | |
| chow | 13.97 | 0 | 0 | 0 | 25.43 | chow | 25.43 | 0 | 0 | 0 | 36.02 |
| diningtable | 11.88 | 21.62 | 18.18 | 0.2 | | diningtable | 8.42 | 0 | 0 | 0 | |
| dog | 31.53 | 75.61 | 10.8 | 0.19 | | dog | 42.67 | 8.54 | 87.5 | 0.16 | |
| horse | 58.4 | 70.77 | 30.87 | 0.43 | | horse | 57.67 | 50.77 | 61.11 | 0.55 | |
| motorbike | 38.39 | 66.22 | 20.25 | 0.31 | | motorbike | 44.53 | 29.73 | 78.57 | 0.43 | |
| person | 58.52 | 84.11 | 16.55 | 0.28 | | person | 59.38 | 75.84 | 33.35 | 0.46 | |
| pottedplant | 19.2 | 40.17 | 21.17 | 0.28 | | pottedplant | 21.05 | 21.37 | 32.57 | 0.26 | |
| sheep | 36.62 | 80.39 | 8.95 | 0.16 | | sheep | 55.23 | 56.86 | 45.31 | 0.5 | |
| sofa | 19.43 | 62.07 | 9.89 | 0.17 | | sofa | 10.18 | 10.34 | 16.67 | 0.13 | |
| train | 24.09 | 42.11 | 18.6 | 0.26 | | train | 38.55 | 57.89 | 36.07 | 0.44 | |
| tvmonitor | 53.74 | 71.74 | 10.48 | 0.18 | | tvmonitor | 27.36 | 36.96 | 15.74 | 0.22 | |

Table 13: Fine-grained analysis of fusion object detection tasks using AutoFusion

| | Setting | Convergence Step(↓) | Optimizable parameters(↓) |
|---|---|---|---|
| Ensemble | CNN + MNIST | 18740 | 567226 |
| | MLP + MNIST | 16900 | 1720330 |
| | CNN + Fashion | 21551 | 567226 |
| | MLP + Fashion | 14992 | 1720330 |
| | CNN + KMNIST | 22488 | 567226 |
| | MLP + KMNIST | 15929 | 1720330 |
| | CNN + CIFAR10 | 37480 | 567226 |
| | MLP + CIFAR10 | 38417 | 1720330 |
| AutoFusion | CNN + MNIST(5+5) | 932 | 267328 |
| | MLP + MNIST(5+5) | 1020 | 802660 |
| | CNN + Fashion(5+5) | 859 | 267328 |
| | MLP + Fashion(5+5) | 800 | 802660 |
| | CNN + KMNIST(5+5) | 937 | 267328 |
| | MLP + KMNIST(5+5) | 792 | 802660 |
| | CNN + CIFAR10(5+5) | 2811 | 267328 |
| | MLP + CIFAR10(5+5) | 1267 | 802660 |

Table 14: Comparison between the learnable parameters and the number of convergence steps of AutoFusion compared to the ensemble training method under different settings.

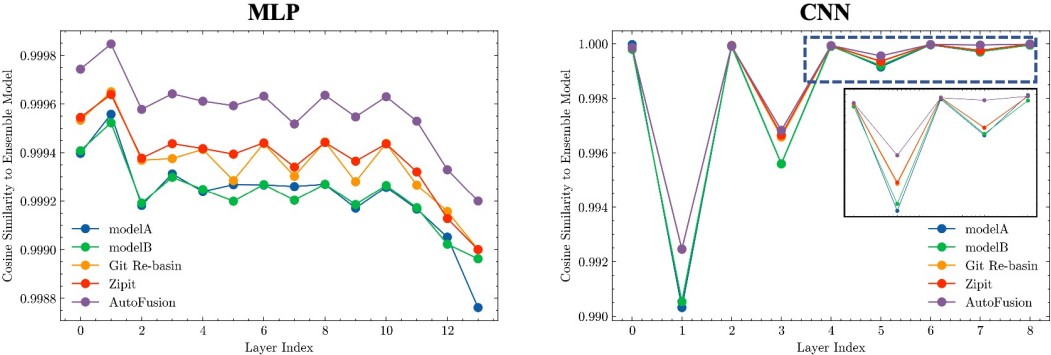

Figure 5: Layer-by-Layer Measure of Similarity of Model Parameters

the integrated training. In terms of optimizable parameters, AutoFusion requires only about half the number of parameters of the ensemble model. This proves that model fusion using AutoFusion has a great advantage in computational performance over training the integrated model on the complete dataset, and the results of the fused model are comparable to those of the integrated model, which further proves that our approach is very promising for exploration.

# F   MORE VISUALIZATION

## F.1   DEGREE OF PARAMETER SIMILARITY TO THE ENSEMBLE MODEL

We calculate the similarity of the parameters of each layer of the fusion model obtained from the different baseline methods by matching the parameters of each layer of the fusion model with the parameters of each layer of the ensemble model one by one, and the one that has a higher similarity with the parameters of each layer of the ensemble model we can consider it as a better fusion method. Specifically, we obtain the parameter vectors of the model by spreading each layer of the model's parameters and restricting it to standard intervals using the Softmax operation, and for two models with the same architecture, we can use cosine similarity to measure the similarity between the layer-by-layer parameter vectors of the two models. The experimental results obtained are shown in Figure 5.

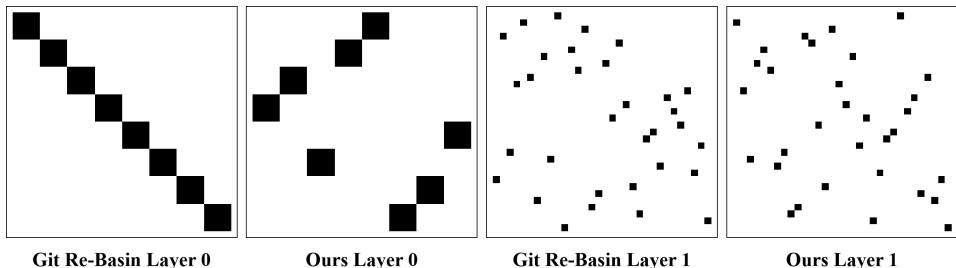

| Git Re-Basin Layer 0 | Ours Layer 0 | Git Re-Basin Layer 1 | Ours Layer 1 |

Figure 6: Export the trained permutation matrix and compare it with Git Re-Basin Method.

It is easy to see from the figure that the fused models obtained by AutoFusion have a higher similarity to the ensemble model at each layer, both for the network with MLP architecture and for the network with CNN architecture, which largely proves that our assumption is well-founded, i.e., *this assumption of high similarity falls apart when the models to be merged are trained for different tasks.*, for models that only use similarity for matching such as Git Re-basin and Zipit, the fused models are not as similar to the ensemble model, while the diversity parameter dedicated to encouraging the AutoFusion method achieved a higher similarity to the ensemble model.

### F.2 Feature Extraction Capability of Fusion Models

**Setting** We conducted some visualizations of the activation maps in the intermediate layers of the multitask model trained on MNIST and the model after fusion, aiming to evaluate the effect of the fused model from a more intuitive perspective. As illustrated in Figure 7, the first column displays the input data. The second column shows the activation map for Model A in response to this input, and the third column presents the activation map for Model B corresponding to the same input (where Model A and Model B are models trained separately on divided datasets, with the specific division details and training methods referred to section 4). Meanwhile, the fourth column depicts the activation map for the output using the AutoFusion method to fuse Models A and B.

**Analysis** It is evident from the figure that, since Model A and Model B were trained only on subsets of the dataset, their ability to extract features is inferior for some inputs. However, the fused model (Fused) clearly demonstrates an improved capability to integrate the feature extraction abilities of different models, capturing key features for all inputs. This provides a more intuitive validation of the effectiveness of the AutoFusion parameter fusion algorithm.

### F.3 Permutation Matrix

We visualize the permutation matrices learned by the Git Re-Basin method and our method (only the first two layers) as shown in Figure 6. By observing the permutation matrix of the first layer, it can be found that our permutation matrix can learn more complex permutations to some extent, whereas the Git Re-Basin

| Layer Index | Git Re-Basin | Ours |
|---|---|---|
| Layer 1 | 64 | 62 |
| Layer 2 | 126 | 128 |
| Layer 3 | 1024 | 1024 |

method in the first layer resembles direct linear interpolation. Starting from the second layer, we calculate the degree of permutation complexity using the $L1$ distance subsection F.3. It can be observed that, given similar levels of permutation complexity, our method achieves better results. This indirectly demonstrates that our method can learn more valuable permutation matrices.

### F.4 An Overview of CAMs

In this section, we provide additional visualizations of model activation maps on the MNIST/KMNIST datasets for reference, aiming to further understand the advantages of the actual fusion model and potential issues that may still exist. The arrangement of rows and columns is consistent with the subsection F.2, with results on the MNIST dataset illustrated in Figure 9, and those on the KMNIST dataset shown in Figure 10.

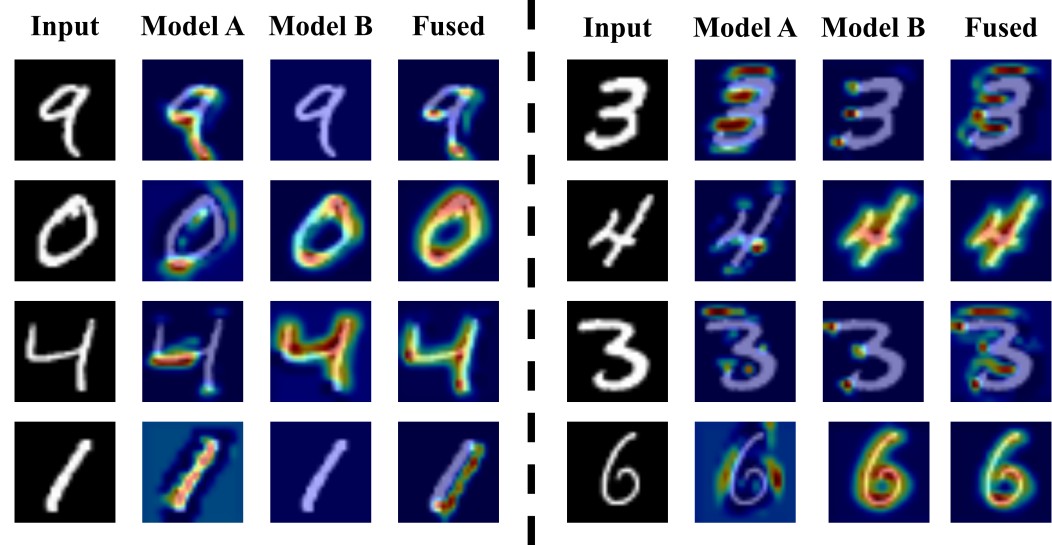

Figure 7: Visualization of the ability of the model to capture features before and after fusion.

It's particularly noteworthy that in these visualizations, we can observe that for some inputs, neither Model A nor Model B is capable of effectively extracting features. However, the model post-AutoFusion integration outperforms both pre-fusion models in terms of feature extraction. This further demonstrates that AutoFusion has learned a more valuable permutation.

Given that the core of the AutoFusion algorithm involves learning the permutation matrix of model parameters, this section provides visualizations of the permutation matrices learned by the Auto-Fusion algorithm when running on three datasets (MNIST, KMNIST, Fashion-MNIST). This offers a more visual display of the AutoFusion method, where the division method for each dataset and the model training process is consistent with section 4. As illustrated, since the CNN(VGG) model we utilized comprises four convolutional layers, our consideration for permutation also exclusively encompasses these four layers.

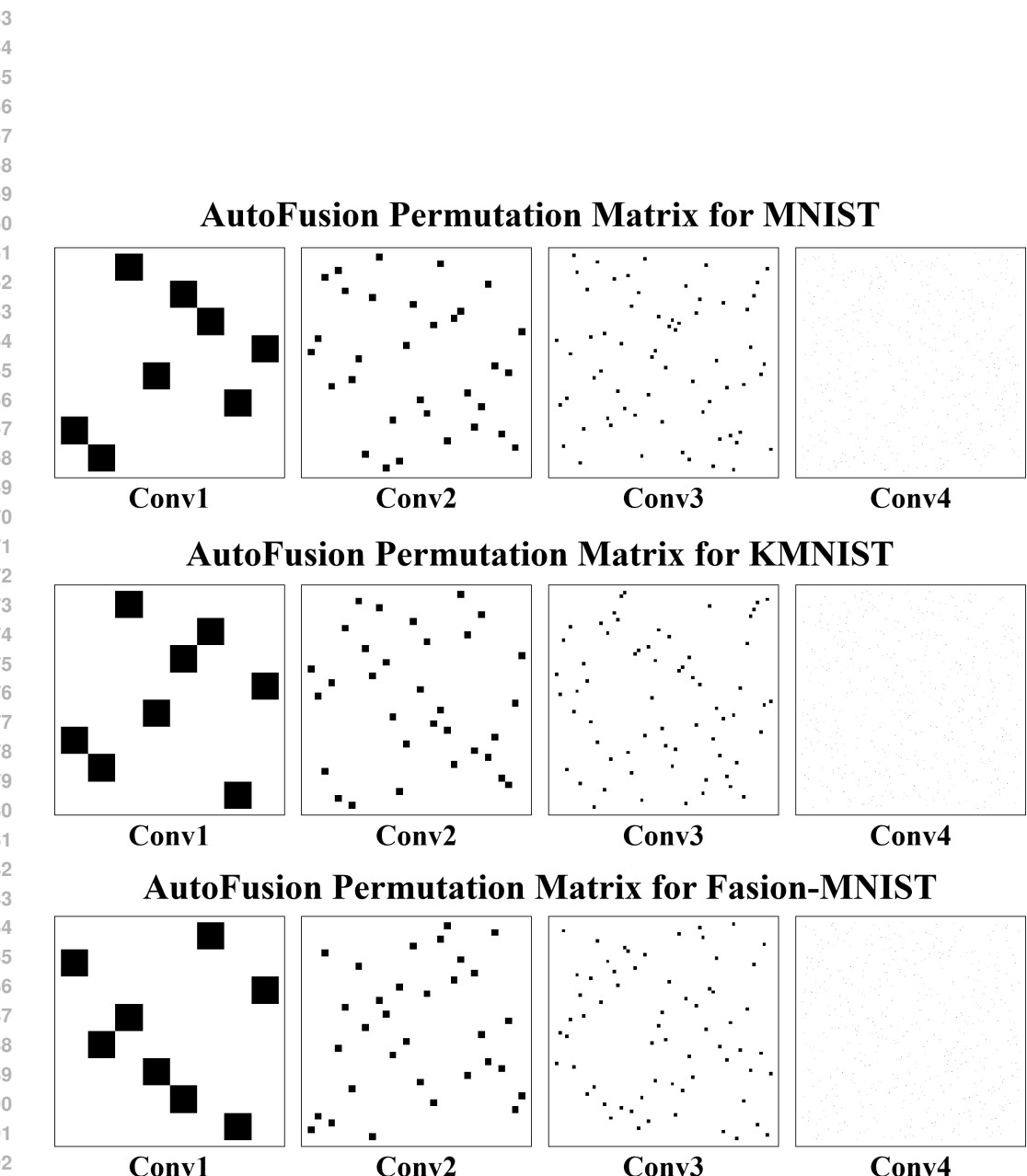

Figure 8: Visualization of permutation matrices.

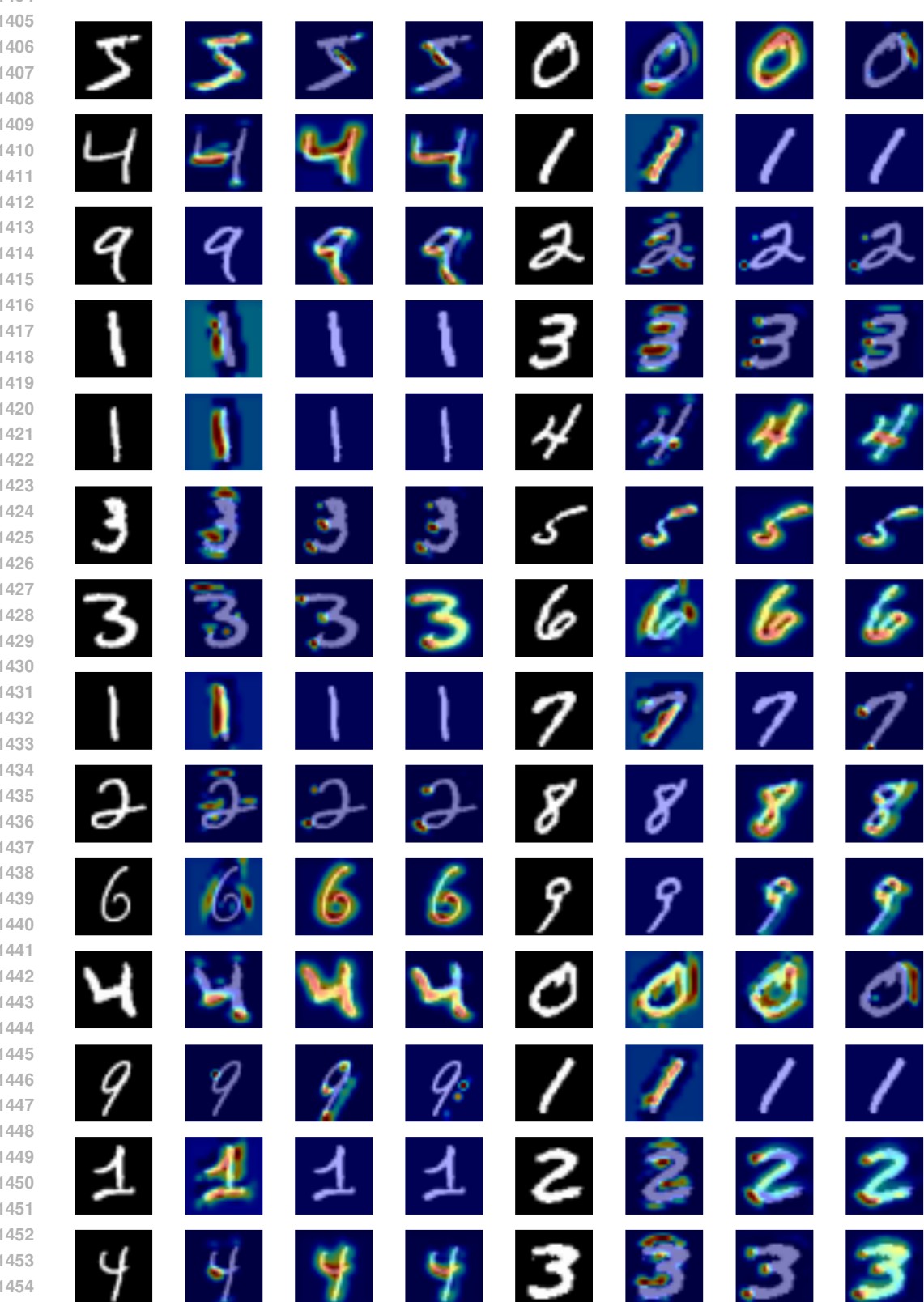

Figure 9: CAMs visualization results on the MNIST dataset.

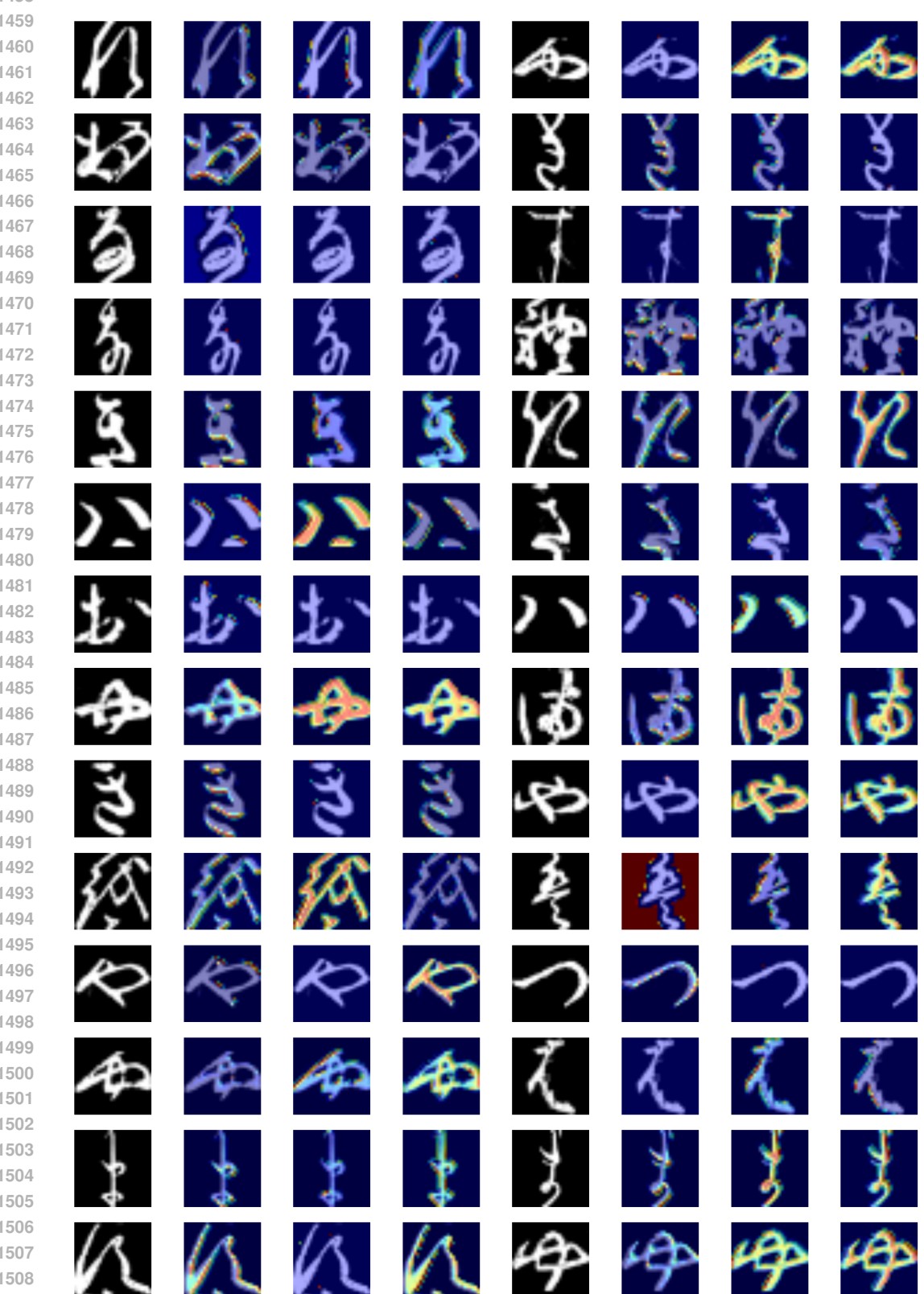

Figure 10: CAMs visualization results on the KMNIST dataset.

