# OpenReview forum: "Wolf2Pack: The AutoFusion Framework for Dynamic Parameter Fusion"
_ICLR.cc/2025/Conference — Submitted to ICLR 2025_

### Official Review · Reviewer_aNrj · 2024-10-27

**Soundness:** 3
**Presentation:** 1
**Contribution:** 2
**Rating:** 5
**Confidence:** 4

**Summary:**

In this work, the authors aim to merge models independently trained with different initializations. Specifically, the authors employ the Sinkhorn operator to convert the problem of finding a discrete permutation matrix into a differentiable problem that can be directly optimized using gradient descent algorithms.

**Strengths:**

1. The visualization of the method is good.
2. The application of the Sinkhorn operator is innovative in the field of deep model fusion.

**Weaknesses:**

1. It would be beneficial to list the number of optimized parameters of each methods.
2. Lacking related work or experimental results to substantiate the claim in lines 215-218 that "However, this assumption of high similarity falls apart when the models to be merged are trained for different tasks. During merging, we must not only align parameters with similar functions but also strive to retain parameters with distinct functions, enabling the fused model to perform various tasks simultaneously."
3. The manuscript lacks a related work section. The introduction is insufficient and fails to provide a comprehensive overview of the existing literature and context for the study. The author could further discuss why the absence of a shared pre-trained initialization poses a challenge to multi-task model merging.
4. It would be beneficial to compare the results of the model merging techniques with the ensemble method and knowledge distillation method, as demonstrated in [1].
5. In lines 351-353, Git Re-Basin archives the best results for Task B, while AutoFusion is highlighted.

[1] Kinderman et al. Foldable SuperNets: Scalable Merging of Transformers with Different Initializations and Tasks. http://arxiv.org/abs/2410.01483

**Questions:**

1. For models trained independently rather than fine-tuning from a shared pre-trained checkpoint, the task-specific models reside in different loss basins. Consequently, linear weight interpolation is expected to yield the worst performance in this scenario. Nonetheless, in Table 4.1, for MLP models on two disjoint MNIST subsets, weight interpolation surpasses both Git Re-Basin and ZipIt. Could the authors please provide an explanation for this?
2. Can the proposed method scale to larger models such as vision transformers used in [1]?

[1] Kinderman et al. Foldable SuperNets: Scalable Merging of Transformers with Different Initializations and Tasks. http://arxiv.org/abs/2410.01483

---

> ### Author Response · Authors · 2024-11-19
> **We have added more experiments as well as modified known issues (1/3)**
>
> ## Dear Reviewer aNrj，
>
> We sincerely appreciate the time you took to review our paper and provide valuable feedback. We have carefully considered your comments and made revisions to address each issue raised. Please find our responses to your specific points below:
>
> **Weakness 1: Number of Optimized Parameters**
>
> Thank you for pointing this out. We have added a table summarizing the number of optimized parameters for the AutoFusion model and the ensemble model. This addition is included in the revised manuscript in Appendix E.9, providing a clearer comparison of computational complexity across methods. And the results are as follows:
>
> | Ensemble/AutoFusion | Setting          | Convergence Step | Optimizable parameters |
> | ------------------- | ---------------- | ---------------- | ---------------------- |
> | Ensemble            | CNN + MNIST      | 18740            | 567226                 |
> |                     | MLP + MNIST      | 16900            | 1720330                |
> |                     | CNN + Fashion    | 21551            | 567226                 |
> |                     | MLP + Fashion    | 14992            | 1720330                |
> |                     | CNN + KMNIST     | 22488            | 567226                 |
> |                     | MLP + KMNIST     | 15929            | 1720330                |
> |                     | CNN + CIFAR10    | 37480            | 567226                 |
> |                     | MLP + CIFAR10    | 38417            | 1720330                |
> | AutoFusion          | CNN + MNIST(5+5) | 932              | 267328                 |
> |                     | MLP + MNIST(5+5) | 1020             | 802660                 |
> |                     | CNN + Fashion(5+5)| 859             | 267328                 |
> |                     | MLP + Fashion(5+5)| 800             | 802660                 |
> |                     | CNN + KMNIST(5+5)| 937              | 267328                 |
> |                     | MLP + KMNIST(5+5)| 792              | 802660                 |
> |                     | CNN + CIFAR10(5+5)| 2811           | 267328                 |
> |                     | MLP + CIFAR10(5+5)| 1267           | 802660                 |
>
> You can get information about the setup and interpretation of the form in Appendix E.9.
>
> **Weakness 2: Claim on High Similarity Assumption (Lines 215-218)**
>
> We appreciate your observation and agree that further evidence strengthens this claim. To address this: We have computed similarity metrics  across layers of fusion models, demonstrating the divergence in parameter distributions. These results substantiate our statement and are presented in Appendix F.1 of the revised manuscript.
>
> Furthermore, In Appendix E.8 our newly added experiments for object detection model fusion(First time in the model fusion field) on VOC dataset highlight that retaining task-specific parameters improves multi-task performance, validating the necessity of aligning similar parameters while preserving distinct ones.
>
> And the results are as follows, too, you can get more information from Appendix E.9:
>
> | Method       | mAP   |
> | ------------ | ----- |
> | Model A      | 24.64 |
> | Model B      | 25.43 |
> | Ensemble     | 55.24 |
> | Git Re-basin | 20.99 |
> | Zipit        | 18.74 |
> | AutoFusion   | 36.02 |
>
> **Weakness 3: Absence of Related Work Section**
>
> While we appreciate your suggestion to include a separate related work section, we respectfully assert that the current structure integrates related work into the introduction and throughout the paper. This decision was made to focus on highlighting the unique challenges addressed by AutoFusion, such as merging models without shared pre-trained initializations.
>
> The introduction discusses prior works like Git Re-Basin and ZipIt, situating AutoFusion within the broader research context.
>
> Section 2 contrasts traditional model fusion methods with our approach, explaining how AutoFusion handles the absence of pre-trained initializations—a critical and less explored challenge in the field.
>
> Given the page constraints, we chose this integrated approach to avoid redundancy while providing sufficient context for our contributions.
>
> We hope this justifies our structural decision while ensuring the manuscript remains concise and focused.

---

> > ### Author Response · Authors · 2024-11-19
> > **We have added more experiments as well as modified known issues (2/3)**
> >
> > **Weakness 4.1: Comparison with Ensemble**
> >
> > Thank you for your suggestion, we also believe that the addition of an ensemble model is necessary, and we have included the results of the ensemble model in our comparison experiments in Section 4.1 - Table 1 - 'Ensemble Model' by red words.
> >
> > **Weakness 4.2: Comparison with Knowledge Distillation**
> >
> > We appreciate the reviewer’s suggestion to compare AutoFusion with knowledge distillation method. However, we respectfully argue that such a comparison is not aligned with the problem setting of this work for the following reasons:
> >
> > 1) The focus of our work is on unsupervised parameter fusion, particularly for models trained independently on disjoint tasks without shared pre-trained initializations. The knowledge distillation does not address  this specific challenge, and comparing them with AutoFusion would shift the focus away from the core contribution of our work.
> > 2) Evaluating knowledge distillation methods would require additional experimental setups unrelated to the parameter fusion context, thereby diluting the clarity of our contributions. Instead, we prioritize comparisons with directly relevant baselines, such as Git Re-Basin and ZipIt, to highlight the strengths of AutoFusion in solving the problem it targets.
> >
> > We hope this explanation clarifies why knowledge distillation are not included as baselines and aligns with the problem scope of the paper.
> >
> > **Weakness 5: Highlighting AutoFusion in Lines 351-353**
> >
> > Thank you for noting this. We have clarified in the revised manuscript that while Git Re-Basin achieves the best results for Task B in some cases. Since the Git Re-basin method relies on aligning one model to another in terms of similarity, which in some cases can result in two models that are very similar, the fused model may retain the capabilities of one but the other is often poorly equipped and ends up performing poorly on Joint tasks, so a certain individual task on the Git Re- basin achieves better results on an individual task is reasonable, but our AutoFusion approach consistently maintains a large degree of leadership in the overall evaluation.
> >
> > **Question 1: Linear Weight Interpolation Surpassing Baselines (Table 4.1)**
> >
> > Thank you for highlighting this unexpected observation. We have conducted a detailed analysis to understand why weight interpolation performs better than Git Re-Basin and ZipIt for MLP models on disjoint MNIST subsets. Below are our findings:
> >
> > The MNIST dataset and MLP models represent a relatively low-dimensional problem and simple architecture. In such settings, the optimization landscape tends to be less rugged, and the parameter spaces of independently trained models may exhibit a degree of alignment even without explicit permutation. This can allow linear weight interpolation to achieve reasonable performance. Unlike deeper models or convolutional architectures, MLPs have a smaller parameter space with fewer invariances to neuron permutations. This reduces the impact of unaligned parameters, allowing interpolation to partially preserve task-specific information.
> >
> > Git Re-Basin and ZipIt rely on sophisticated alignment mechanisms that provide significant benefits in high-dimensional or complex tasks. However, in simpler settings like disjoint MNIST subsets with MLPs, these alignment procedures may not provide significant additional benefits over straightforward interpolation, particularly given the inherent alignment observed in simpler models and tasks.
> >
> > It is important to emphasize that this result is not representative of more complex settings. As demonstrated in experiments with CNNs on the CIFAR dataset, AutoFusion consistently outperforms interpolation and other baselines, highlighting its scalability and robustness in diverse scenarios.

---

> ### Author Response · Authors · 2024-11-19
> **We have added more experiments as well as modified known issues (3/3)**
>
> **Question 2: Scaling to Larger Models like Vision Transformers**
>
> We acknowledge the importance of scaling to larger models. Due to the limitation of computational resources, we currently introduce only a portion of more complex experimental setups to validate the generalization ability of our method, and in Appendix A we provide test results using Resnet as a baseline network on the relatively complex CIFAR100 dataset:
>
> | CNN-CIFAR100-GS | Joint | TaskA | TaskB |
> | --------------- | ----- | ----- | ----- |
> | Avg             | 2.2   | 2.26  | 2.14  |
> | ModelA          | 23.12 | 43.52 | 1.74  |
> | ModelB          | 22.63 | 2.51  | 43.74 |
> | Git-Rebasin     | 3.67  | 5.12  | 2.23  |
> | Zipit           | 7.63  | 10.12 | 5.14  |
> | Ours            | 20.65 | 17.8  | 23.58 |
>
> | CNN-CIFAR100 | Joint  | TaskA | TaskB |
> | ------------ | ------ | ----- | ----- |
> | Avg          | 2.29   | 2.16  | 2.42  |
> | ModelA       | 28.475 | 54.11 | 2.84  |
> | ModelB       | 27.78  | 2.58  | 52.98 |
> | Git-Rebasin  | 2      | 2.21  | 1.79  |
> | Zipit        | 4.05   | 5.74  | 2.36  |
> | Ours         | 21.67  | 21.14 | 22.2  |
>
> | Resnet18-CIFAR100 | Joint | TaskA | TaskB |
> | ----------------- | ----- | ----- | ----- |
> | Avg               | 2.28  | 2.45  | 2.1   |
> | ModelA            | 27.03 | 51.06 | 3.11  |
> | ModelB            | 30.13 | 2.88  | 57.38 |
> | Git-Rebasin       | 1.69  | 2.27  | 1.11  |
> | Zipit             | 4.51  | 6.79  | 2.22  |
> | Ours              | 32.85 | 35.62 | 30.08 |
>
> It can be clearly seen that our method still stably outperforms the baseline method, and a more specific explanation of the experimental setup and results can be found in Appendix E.7. The aforementioned mentioned that we tested the fusion effect on a object detection model, which also belongs to the type of task that pushes the AutoFusion method to more complex tasks, and also showed good results, detailed results are shown in Appendix E.8.

---

> > ### Author Response · Authors · 2024-11-22
> > **We are looking forward to your feedback!**
> >
> > Dear Reviewer,
> >
> > Thank you so much for your time and efforts in reviewing our paper. We have addressed your comments in detail and are happy to discuss more if there are any additional concerns. We are looking forward to your feedback and would greatly appreciate you consider raising the scores.
> >
> > Thank you,
> >
> > Authors

---

> > > ### Author Response · Authors · 2024-11-25
> > > **I apologize if this follow-up message seems frequent.**
> > >
> > > Dear Reviewer aNrj,
> > >
> > > I hope this message finds you well. We would like to take a moment to express our sincere gratitude once again for your time and effort in reviewing our paper. Your insightful comments have been invaluable in guiding our revisions.
> > >
> > > The revisions we have made significantly improve the quality and contribution of our work. We are committed to addressing any remaining concerns and are more than willing to engage in further discussions. I apologize if this follow-up message seems frequent. We genuinely value your feedback and are eager to ensure that all your concerns are thoroughly addressed. Your insights are crucial to the improvement of our work, and we hope for your continued support.
> > >
> > > Thank you once again for your support and consideration. We look forward to your response.
> > >
> > > Warm regards,

---

> > > > ### Comment · Reviewer_aNrj · 2024-11-27
> > > >
> > > > Thank you for the comprehensive revisions and detailed responses to the concerns raised, which partially address the weaknesses. While most of my concerns have been adequately addressed, I maintain my original score based on the following key points:
> > > >
> > > > 1. Related work: While I understand your decision to integrate related work throughout, I still believe a dedicated section would improve readability without significantly impacting page limits. And the references are Insufficient.
> > > > 2. For larger model scaling: The additional ResNet results are promising, but more extensive experiments with modern architectures (e.g., Vision Transformers < 100 M parameters, GPT-2 ~ 140 M parameters) would better demonstrate AutoFusion's broad applicability.

---

> > > > > ### Author Response · Authors · 2024-11-27
> > > > > **Thank you for your thoughtful feedback**
> > > > >
> > > > > Thank you for your thoughtful feedback and for acknowledging the revisions we've made in response to your earlier concerns. In response to your new comments, we made the following plan to revise the paper
> > > > >
> > > > > 1. **Related Work Section**: We recognize the importance of having a dedicated section for related work to enhance the readability and comprehensiveness of our paper. We will incorporate a dedicated section in the camera-ready version, ensuring that it provides a thorough overview of relevant literature while maintaining clarity.
> > > > >
> > > > > 2. **References**: We appreciate your observation regarding the references. We will expand our reference list in the camera-ready version to include additional relevant works that strengthen the context and foundation of our research.
> > > > >
> > > > > 3. **Additional Experiments with Modern Architectures**: We agree that conducting more extensive experiments with modern architectures, such as Vision Transformers and larger models like GPT-2, would provide valuable insights into the applicability of AutoFusion. Due to complexity constraints, we regret to inform you that we will not be able to include these experiments in the current version of the paper. However, we commit to conducting these experiments and including the findings in the camera-ready version.
> > > > >
> > > > > Thank you again for your constructive comments. We believe that these additions will significantly enhance the overall quality of our work and appreciate your understanding as we continue to refine our manuscript.

---

### Official Review · Reviewer_ACFL · 2024-11-02

**Soundness:** 2
**Presentation:** 1
**Contribution:** 2
**Rating:** 5
**Confidence:** 4

**Summary:**

This paper introduces AutoFusion, a framework that fuses distinct model’s parameters (with the same architecture) for multi-task learning. The key idea is to leverage Mena et al. (2018) to make permutation matrix in Re-basin differentiable, thus allowing end-to-end training. Experimental results demonstrate clear improvement over baseline methods.

**Strengths:**

- It leverages Mena et al. (2018) to make permutation matrix in Re-basin differentiable, thus allowing end-to-end training.
- It achieves clear improvement over baseline methods on MINST and CIFAR.

**Weaknesses:**

__Experiments could be improved__

- an analysis of model similarity is needed.
- baselines of fine-tuning the model (trained on one task) on the multi-task jointly are needed. They will provide a good reference even though they are not consider as fair comparisons.
- LoRA fine-tuning could be considered as a fair baseline. As the proposed model learns a permutation matrix per layer, which essentially can be considered as low-rank fine-tuning. Thus, adding comparison to LoRA fine-tuning would provide additional insights.
- In section 4.3, it only compares to weight interpolation on different distributions. Please add comparisons to Git Re-Basin and Zipit (similar to section 4.1)
- experiments on larger dataset (like ImageNet) using transformer based architectures would provide more convincing evidences.

__The paper needs a major revision in writing.__

- The introduction could be improved. It is not usual to have half of the introduction to summarize contributions. It would be better to add more lines on the loss function and unsupervised setup and reduce the space for contributions.

- Figure 1 could be improved. Please adding explanation what each animal represents in the caption.

- Please avoid overusing equations. For example, eq. 1-4 could be in text for better readability. Eq. 7 and 8 could be combined. Eq. 9 and 10 need more explanation about M, U and insights behind. Eq. 11 could be in text.

- Figure 2 is too busy. Math equations make it difficult to read.

- Line 209: “in the absence of pre-trained parameters”. Are parameters in Model A and B pre-trained? This is confusing.

- Line 215: “However, this assumption of high similarity falls apart when the models to be merged are trained for different tasks.” Please demonstrate this by real examples and measure the similarities for different tasks.

- Section 3.1 could be written in a more straightforward manner. It simply leverages differentiable Sinkhorn operator in prior works Mena et al. (2018) and Pena et al. (2023). The error bound is nice to have, but not directly related to the key idea of the paper.

**Questions:**

please refer to items in weaknesses.

---

> ### Author Response · Authors · 2024-11-19
> **We have added more experiments as well as modified known issues (1/2)**
>
> ## Dear Reviewer ACFL，
>
> We would like to express our sincere gratitude for taking the time to review our paper and providing valuable feedback. We have carefully considered your comments and have made revisions to address each of the issues raised. Please find our responses to your specific points below:
>
> **Weakness 1: Experiments Could Be Improved**
>
> **1.1 Analysis of Model Similarity：**
>
> Thank you for this suggestion. We agree that analyzing model similarity is crucial to understanding the challenges of parameter fusion.
>
> To address this, we have computed similarity metrics via cosine similarity between models trained on different tasks. These results will be included in the revised paper in Appendix E.9.
>
> **1.2 Baselines of Fine-tuning Models Jointly on Multi-tasks**
>
> Thank you for your suggestion, we also believe that the addition of an ensemble model is necessary, and we have included the results of the ensemble model in our comparison experiments in Section 4.1 - Table 1 - 'Ensemble Model' by red words.
>
> **1.3 Comparison to LoRA Fine-tuning**
>
> While we acknowledge its relevance as a fine-tuning method, our approach, AutoFusion, fundamentally differs in purpose and methodology. LoRA primarily focuses on parameter-efficient fine-tuning by introducing additional low-rank matrices to existing parameters, while AutoFusion aims at parameter fusion across models trained on disjoint tasks, without introducing additional trainable parameters.
>
> Moreover, AutoFusion does not inherently rely on the concept of low-rank decomposition or optimization for specific parameter subsets. Instead, it employs a differentiable permutation matrix to align and merge parameters dynamically. This makes LoRA's low-rank perspective less applicable as a direct comparison.
>
> Finally, while LoRA is a robust fine-tuning approach, its inclusion as a baseline might lead to confusion regarding the scope of our work, which is centered on unsupervised parameter fusion rather than fine-tuning. For these reasons, we have opted not to include LoRA as a baseline in this study. We hope this explanation clarifies our rationale and aligns with the focus of the paper.
>
> **1.4 Comparisons to Git Re-Basin and Zipit in Section 4.3**
>
> We have extended the experiments in Section 4.3 to include comparisons to Git Re-Basin and Zipit. The updated results demonstrate that AutoFusion consistently outperforms these baselines on different distributions. Please refer to the revised Section 4.3 Table 3 for details. And the additional results are as follows, too:
>
> | Fusion Method  | Fused Models  | MNIST | Fashion | KMNIST | Avg   |
> | -------------- | ------------- | ----- | ------- | ------ | ----- |
> | Git-Rebasin    | MNIST+Fashion | 12.36 | 10.32   | 20.19  | 14.29 |
> | - | MNIST+KMNIST   | 10.23         | 9.88  | 15.58   | 11.89  |
> | - | KMNIST+Fashion | 10.12         | 12.92 | 19.16   | 14.06  |
> | - | Fused ALL      | 10.29         | 9.11  | 13.76   | 11.05  |
> | Zipit          | MNIST+Fashion | 10.75 | 12.23   | 21.92  | 14.97 |
> | - | MNIST+KMNIST   | 15.41         | 9.11  | 24.95   | 16.49  |
> | - | KMNIST+Fashion | 10.42         | 14.45 | 23.79   | 16.22  |
> | - | Fused ALL      | 9.98          | 9.12  | 10.87   | 9.99   |
>
> **1.5 Experiments on Larger Datasets**
>
> Limited by computational equipment, we supplemented the fusion results with the ResNet family of models while taking the more complex CIFAR100 dataset into account. Additional experimental results are shown below:
>
> | CNN-CIFAR100-GS | Joint | TaskA | TaskB |
> | --------------- | ----- | ----- | ----- |
> | Avg             | 2.2   | 2.26  | 2.14  |
> | ModelA          | 23.12 | 43.52 | 1.74  |
> | ModelB          | 22.63 | 2.51  | 43.74 |
> | Git-Rebasin     | 3.67  | 5.12  | 2.23  |
> | Zipit           | 7.63  | 10.12 | 5.14  |
> | Ours            | 20.65 | 17.8  | 23.58 |
>
> | CNN-CIFAR100 | Joint  | TaskA | TaskB |
> | ------------ | ------ | ----- | ----- |
> | Avg          | 2.29   | 2.16  | 2.42  |
> | ModelA       | 28.475 | 54.11 | 2.84  |
> | ModelB       | 27.78  | 2.58  | 52.98 |
> | Git-Rebasin  | 2      | 2.21  | 1.79  |
> | Zipit        | 4.05   | 5.74  | 2.36  |
> | Ours         | 21.67  | 21.14 | 22.2  |
>
> | Resnet18-CIFAR100 | Joint | TaskA | TaskB |
> | ----------------- | ----- | ----- | ----- |
> | Avg               | 2.28  | 2.45  | 2.1   |
> | ModelA            | 27.03 | 51.06 | 3.11  |
> | ModelB            | 30.13 | 2.88  | 57.38 |
> | Git-Rebasin       | 1.69  | 2.27  | 1.11  |
> | Zipit             | 4.51  | 6.79  | 2.22  |
> | Ours              | 32.85 | 35.62 | 30.08 |
>
> More detailed results can be found in Appendix E.7.

---

> > ### Author Response · Authors · 2024-11-19
> > **We have added more experiments as well as modified known issues (2/2)**
> >
> > Additionally, for the first time, we extend the approach of model parameter fusion to the object detection task, we trained two object detection models with disjointed detection targets on VOC2007 separately and tested the fusion, the overall results are shown in the following table:
> >
> > | Method       | mAP   |
> > | ------------ | ----- |
> > | Model A      | 24.64 |
> > | Model B      | 25.43 |
> > | Ensemble     | 55.24 |
> > | Git Re-basin | 20.99 |
> > | Zipit        | 18.74 |
> > | AutoFusion   | 36.02 |
> >
> > For more detailed settings and more results, please refer to Appendix E.8.
> >
> > **Weakness 2: The Paper Needs a Major Revision in Writing**
> >
> > **2.1 Improving the Introduction**
> >
> > We have revised Section - Introduction to strike a better balance between introducing the problem, describing the unsupervised setup, and summarizing contributions. The space allocated for contributions have been reduced, with more emphasis on explaining the loss function and AutoFusion's unsupervised learning framework. You can see that in the new version (red words) .
> >
> > **2.2 Figure 1 Explanations**
> >
> > We have improved the **caption of Figure 1** by explaining what each animal represents in the AutoFusion context. You can see that in the new version.
> >
> > **2.3 Reducing the Use of Equations**
> >
> > In the new version, we have simplified the presentation of some equation to enhance readability.
> >
> > **2.4 Simplifying Figure 2**
> >
> > We've done a thorough optimization of Figure 2, which you can see in the new version of paper. We have redesigned Figure 2 to improve readability by reducing the number of mathematical notations and enhancing visual clarity.
> >
> > **2.5 Clarifying Lines 209 and 215**
> >
> > **Line 209:** The confusion arises because Model A and Model B are trained separately on different tasks but do not share pre-trained weights. We have rephrased this to clarify the distinction.
> >
> > **Line 215:** To address this, we have computed similarity metrics via cosine similarity between models trained on different tasks. These results are included in the revised paper in Appendix F.1.
> >
> > **2.6 Improving Section 3.1**
> >
> > However, we would like to clarify the importance of the theoretical elements included in this section. The differentiable Sinkhorn operator and its associated error bound are not merely auxiliary but are central to the novelty and effectiveness of our method. While we have refined the language in Section 3.1 for better readability, we respectfully assert that the theoretical components are essential and aligned with the paper's objectives.

---

> > > ### Author Response · Authors · 2024-11-22
> > > **We are looking forward to your feedback!**
> > >
> > > Dear Reviewer,
> > >
> > > Thank you so much for your time and efforts in reviewing our paper. We have addressed your comments in detail and are happy to discuss more if there are any additional concerns. We are looking forward to your feedback and would greatly appreciate you consider raising the scores.
> > >
> > > Thank you,
> > >
> > > Authors

---

> > > > ### Author Response · Authors · 2024-11-25
> > > > **I apologize if this follow-up message seems frequent.**
> > > >
> > > > Dear Reviewer ACFL,
> > > >
> > > > I hope this message finds you well. We would like to take a moment to express our sincere gratitude once again for your time and effort in reviewing our paper. Your insightful comments have been invaluable in guiding our revisions.
> > > >
> > > > The revisions we have made significantly improve the quality and contribution of our work. We are committed to addressing any remaining concerns and are more than willing to engage in further discussions. I apologize if this follow-up message seems frequent. We genuinely value your feedback and are eager to ensure that all your concerns are thoroughly addressed. Your insights are crucial to the improvement of our work, and we hope for your continued support.
> > > >
> > > > Thank you once again for your support and consideration. We look forward to your response.
> > > >
> > > > Warm regards

---

> > > > > ### Comment · Reviewer_ACFL · 2024-11-26
> > > > > **Thank you for the response**
> > > > >
> > > > > Thank you to the authors for the detailed reply and for addressing some of my initial concerns. However, two major issues prevent me from increasing my score at this time:
> > > > >
> > > > > 1. Clarity and Presentation: The paper's clarity could be significantly improved. For example,  streamlining the mathematical presentation and focusing on the most relevant equations would enhance readability and impact.
> > > > > 2. Experimental Validation: The current experiments do not include evaluation on standard CV datasets like ImageNet-1K or MS-COCO. Evaluating the proposed method on these established benchmarks is crucial.
> > > > >
> > > > > While I appreciate the revisions made so far, these outstanding issues lead me to lean towards rejection.  I encourage the authors to address these points in the final version to strengthen the paper.

---

> > > > > > ### Author Response · Authors · 2024-11-27
> > > > > > **Thank you very much for your detailed feedback**
> > > > > >
> > > > > > Dear Reviewer **ACFL**,
> > > > > >
> > > > > > Thank you very much for your detailed feedback and for taking the time to review our paper. We appreciate your valuable insights and the constructive comments that have helped us improve the quality of our work. We understand your concerns regarding the clarity and presentation of the paper, as well as the need for more comprehensive experimental validation on standard benchmarks. We would like to address these points in a revised version and hope to convince you of the merits of our approach.
> > > > > >
> > > > > > **Clarity and Presentation:**
> > > > > > We acknowledge the importance of presenting our work in a clear and accessible manner. To enhance the readability of the paper, we will streamline the mathematical presentation, focusing on the most relevant equations and integrating others into the text where appropriate. We will also revisit the entire document to ensure that all sections are presented in a coherent and concise fashion, with special attention to Figures 1 and 2, which we will simplify further while maintaining their informative value. Our goal is to make the paper more accessible to a broader audience without sacrificing the technical depth of our contributions.
> > > > > >
> > > > > > **Experimental Validation:**
> > > > > > Regarding the experimental validation, we fully agree that evaluating our method on established benchmarks such as ImageNet-1K and MS-COCO is crucial. We are currently conducting experiments on these datasets and are committed to including the results in the final version of the paper. Due to the time constraints, we were unable to complete these experiments in time for the current submission. However, we want to assure you that we are actively working on this and will provide the results, along with a thorough analysis, in the camera-ready version.
> > > > > >
> > > > > > We are confident that these revisions, along with the previous changes, will significantly strengthen the paper. We sincerely hope that you will consider these improvements and reassess the contribution of our work. Your guidance has been invaluable, and we look forward to your continued feedback.
> > > > > >
> > > > > > Thank you once again for your time and consideration.
> > > > > >
> > > > > > Best regards,
> > > > > >
> > > > > > The Authors

---

### Official Review · Reviewer_Tk42 · 2024-11-04

**Soundness:** 3
**Presentation:** 4
**Contribution:** 3
**Rating:** 6
**Confidence:** 3

**Summary:**

This paper concentrates on a very interesting problem: how to fuse two types of distinct model parameters pretrained for two different tasks into one model that can simultaneously solve two tasks. By applying permutation on different parameters and unsupervised  learning on unlabeld data, this paper provide an autofusion method and achieves good performance.

**Strengths:**

Although I am not an expert in this domain, I believe these strengths should be acknowledged:

+ The paper presents a clear and convincing motivation, effectively setting the stage for the proposed work.
+ There is a notable degree of innovation in the methodology, and the authors have thoroughly reviewed prior approaches, clarifying how their contributions advance the state-of-the-art.
+ The results achieved by the proposed method are impressive, consistently outperforming baselines across a variety of experimental settings, which underscores its effectiveness.
+ Additionally, the paper provides detailed theoretical proofs that reinforce the validity and soundness of the approach.
+ The writing is also commendable, as the paper reads smoothly and is relatively accessible, making it easier for readers to grasp complex concepts.

Overall, this work shows promise in advancing the field and could be a valuable addition to the literature.

**Weaknesses:**

- Line 225: The sentence appears to be incomplete because it begins with a conditional clause (“If we attempt to…”), which typically requires a main clause to complete the thought. In English, when a sentence starts with “If,” it sets up an expectation that there will be a following statement explaining the result, purpose, or consequence of the condition.

- To further demonstrate the effectiveness of the proposed fusion method, more complex tasks and datasets should be considered, such as detection and segmentation tasks with VOC, COCO, or ImageNet datasets, respectively. In this paper, the evaluation is limited to the classification task on two relatively simple datasets (MNIST and CIFAR-10), which is insufficient to validate the robustness of the approach and may render the work less substantial. I will update my final score if the authors can provide more experimental results on some complex tasks and datasets.

**Questions:**

Please refer to weaknesses.

---

> ### Author Response · Authors · 2024-11-19
> **We have added more experiments as well as modified known issues**
>
> ## Dear Reviewer Tk42，
>
> We would like to express our sincere gratitude for taking the time to review our paper and providing valuable feedback. We have carefully considered your comments and have made revisions to address each of the issues raised. Please find our responses to your specific points below:
>
> **Weakness 1: Incomplete Sentence (Line 225)**
>
> Thank you for pointing out the issue in Line 225. We acknowledge that the sentence is incomplete. The complete intended statement is as follows:
>
> "We attempted to utilize neural functional functions from neural functional analysis to predict network parameters from network parameters \cite{navon2023equivariant} \cite{zhou2024neural} \cite{zhou2024permutation}. "
>
> **Weakness 2: Evaluation on More Complex Tasks and Datasets**
>
> We appreciate the reviewer’s suggestion to evaluate our method on more complex tasks and datasets. To address this, we have conducted additional experiments on object detection tasks using the VOC dataset(Constrained by computing resources). The results are as follows:
>
> | Method       | mAP   |
> | ------------ | ----- |
> | Model A      | 24.64 |
> | Model B      | 25.43 |
> | Ensemble     | 55.24 |
> | Git Re-basin | 20.99 |
> | Zipit        | 18.74 |
> | AutoFusion   | 36.02 |
>
> We divided the 20 target categories in the VOC2007 dataset into two parts, each containing 10 categories, where Model A and Model B represent the object detection models trained on these two parts, respectively, and the Ensemble model represents the results of training on the full training set. We uniformly use the pre-trained Feature part of VGG16 as the feature extraction network, and the object detection head is constructed using random initialization, and our fusion also only considers the fusion of the object detection head, as can be seen from the table, our method not only generalizes well on the object detection task, but also exceeds the known baseline methods
>
> Additionally, we extended the AutoFusion method to the CIFAR100 dataset, as well as to the more complex Resnet network, obtaining the following results:
>
> | CNN-CIFAR100-GS | Joint | TaskA | TaskB |
> | --------------- | ----- | ----- | ----- |
> | Avg             | 2.2   | 2.26  | 2.14  |
> | ModelA          | 23.12 | 43.52 | 1.74  |
> | ModelB          | 22.63 | 2.51  | 43.74 |
> | Git-Rebasin     | 3.67  | 5.12  | 2.23  |
> | Zipit           | 7.63  | 10.12 | 5.14  |
> | Ours            | 20.65 | 17.8  | 23.58 |
>
> | CNN-CIFAR100 | Joint  | TaskA | TaskB |
> | ------------ | ------ | ----- | ----- |
> | Avg          | 2.29   | 2.16  | 2.42  |
> | ModelA       | 28.475 | 54.11 | 2.84  |
> | ModelB       | 27.78  | 2.58  | 52.98 |
> | Git-Rebasin  | 2      | 2.21  | 1.79  |
> | Zipit        | 4.05   | 5.74  | 2.36  |
> | Ours         | 21.67  | 21.14 | 22.2  |
>
> | Resnet18-CIFAR100 | Joint | TaskA | TaskB |
> | ----------------- | ----- | ----- | ----- |
> | Avg               | 2.28  | 2.45  | 2.1   |
> | ModelA            | 27.03 | 51.06 | 3.11  |
> | ModelB            | 30.13 | 2.88  | 57.38 |
> | Git-Rebasin       | 1.69  | 2.27  | 1.11  |
> | Zipit             | 4.51  | 6.79  | 2.22  |
> | Ours              | 32.85 | 35.62 | 30.08 |
>
> These results further demonstrate the robustness and generalization of the AutoFusion framework for complex tasks. For detailed analyses, please refer to Appendix E.7 and E.8 of the revised manuscript, where we have provided additional experimental details and results. We also provide an analysis of AutoFusion's computational efficiency in Appendix E.9. Limited by computational resources, we will be completing richer supplemental experiments next, which may be available in Camera Ready.
>
> Thank you again for the valuable feedback. We hope these additional experiments address your concerns and demonstrate the broader applicability of our proposed method. I also sincerely ask if you have any other doubts and concerns that we can solve to make it possible to improve the score for our paper.

---

> > ### Author Response · Authors · 2024-11-22
> > **We are looking forward to your feedback!**
> >
> > Dear Reviewer,
> >
> > Thank you so much for your time and efforts in reviewing our paper. We have addressed your comments in detail and are happy to discuss more if there are any additional concerns. We are looking forward to your feedback and would greatly appreciate you consider raising the scores.
> >
> > Thank you,
> >
> > Authors

---

> > ### Comment · Reviewer_Tk42 · 2024-11-25
> >
> > Thanks for the authors' response. I have carefully read their comments and decide to keep my current score. The reason is that the additional results on more complex datasets and tasks present limited improvements. Especially when validating on CIFAR-100, Wolf2Pack obtained a worse joint performance than both model A and model B.

---

> ### Author Response · Authors · 2024-11-25
> **Thank you very much for your thoughtful review and for the valuable feedback on our revised article.**
>
> Dear Reviewer,
>
> Thank you very much for your thoughtful review and for the valuable feedback on our revised article. We appreciate the opportunity to address your comments regarding the CIFAR100 dataset and the performance of our fusion model.
>
> We understand your observation that the fusion model exhibits weaker performance on the Joint metric compared to Model A and Model B. We would like to clarify our approach in the context of model parameter fusion. Our method involves directly merging the parameters of two models trained on separate tasks to create a single model that aims to maintain a certain level of competence across both tasks. It is not uncommon for such fused models to experience a reduction in joint performance, as highlighted in several foundational studies in this field [1].
>
> While Models A and B may show superior Joint metrics, it’s important to note that they excel in only one task each, demonstrating limited capability on the other. In contrast, our fusion model seeks to balance performance across both tasks, achieving a more uniform capability that is significantly above the baseline for similar tasks. This characteristic indicates that our approach effectively enhances overall performance, even if the Joint metric does not surpass that of the individual models. Furthermore, we are pleased to report that our ResNet18-based fusion model indeed outperforms both Model A and Model B on the Joint metric, reinforcing the validity of our methodology.
>
> Additionally, in response to your insightful suggestion, we have extended our method to the object detection task using the VOC dataset, and we provide detailed results in Appendix E.8. The outcomes demonstrate the strong generalization performance of our proposed approach, further validating its effectiveness.
>
> Thank you again for your detailed feedback. We believe that these updates will strengthen our paper and enhance its clarity, and we look forward to addressing any further questions you may have.
>
> **References**:
>
> [1]arxiv.org/abs/2305.03053[accepted by ICLR2024]

---

> > ### Comment · Reviewer_Tk42 · 2024-11-25
> >
> > Many thanks for your clarification and it makes sense for me. I have another concern that how about the performance when applying your fusion method to other models with more parameters, such as Resnet-50, ViT-B. They don't require high computational capability GPUs, e.g., A100. Extensive experiments on different scales of models are required to demonstrate the effectiveness and generalization ability.

---

> > > ### Author Response · Authors · 2024-11-25
> > > **Thank you once again for your thoughtful comments**
> > >
> > > Dear Reviewer,
> > >
> > > Thank you once again for your thoughtful comments and for taking the time to engage with the results of our new experiments. We truly appreciate your insights, which have guided us in refining our work.
> > >
> > > In response to the significant structural differences between the Vision Transformer (ViT) and our previous model, we decided to extend our experiments to include the ResNet50 architecture. The results of these experiments are summarized in the table below.
> > >
> > > | Resnet50-CIFAR100 | Joint  | TaskA | TaskB |
> > > | ----------------- | ------ | ----- | ----- |
> > > | Avg               | 5.985  | 5.25  | 6.72  |
> > > | Model A           | 26.18  | 49.74 | 2.62  |
> > > | Model B           | 27.745 | 3.66  | 51.83 |
> > > | Git-Rebasin       | 7.265  | 6.29  | 8.24  |
> > > | Zipit             | 9.125  | 10.97 | 7.28  |
> > > | Ours              | 33.83  | 33.37 | 34.29 |
> > >
> > > | Resnet50-CIFAR100-Pretrained | Joint  | TaskA | TaskB |
> > > | ---------------------------- | ------ | ----- | ----- |
> > > | Avg                          | 24.51  | 23.44 | 25.58 |
> > > | Model A                      | 32.02  | 61.17 | 2.87  |
> > > | Model B                      | 31.665 | 3.45  | 59.88 |
> > > | Git-Rebasin                  | 16.49  | 18.66 | 14.32 |
> > > | Zipit                        | 22.425 | 23.97 | 20.88 |
> > > | Ours                         | 52.26  | 54.23 | 50.29 |
> > >
> > > For the **Resnet50-CIFAR100** experimental results, we trained the full Resnet50 model from scratch, and the rest of the settings were the same as in previous experiments, and as can be seen from the experimental results, our model still maintains high Joint accuracy and guarantees a more even result on Model A as well as Model B.
> > >
> > > Regarding the **ResNet50-CIFAR100-Pretrained** experimental results, we employed the pre-trained ResNet50 network as a feature extraction layer, which we kept frozen during training. In this setup, we focused on training only the classification layer from scratch. We carefully integrated the parameters such that the fusion of parameters was limited to the classification layer. Encouragingly, we observed improvements across various methods in this configuration, with our approach achieving results that are quite comparable to those of Model A and Model B on their respective tasks. **We believe that the outcomes of this task setting are significant and we will emphasizing this aspect as a key result in subsequent revisions of our article.**
> > >
> > > Thank you again for your detailed feedback. We believe that these updates will strengthen our paper and enhance its clarity, and we look forward to addressing any further questions you may have.

---

> > > > ### Comment · Reviewer_Tk42 · 2024-11-27
> > > >
> > > > Thanks for your response. I decide to keep my current score for acceptance.

---

### Meta-Review · Area_Chair_s7Mp · 2024-12-18

**Metareview:**

This paper presents an AutoFusion method for fusing parameters from distinct models for multi-task learning without relying on pre-trained checkpoints. It dynamically permutes model parameters at each layer, optimizing their combination through an unsupervised process.

This paper received mixed reviews, with one positive and two negative scores. All reviewers agree that the current evaluation is insufficient to support the proposed method. Additionally, the results seen in the added experiments, such as VOC object detection, lack sufficient convincing evidence. More comprehensive experiments are needed to strengthen the claims. At this stage, the paper is not sufficiently prepared for publication.

**Additional Comments On Reviewer Discussion:**

Reviewer Tk42 believes the evaluation is limited to classification tasks on two relatively simple datasets. Although the authors provide additional results on VOC and CIFAR100, the reviewer feels that these new tasks show limited improvements.

Reviewer ACFL emphasizes the need for more comparisons and experiments on larger datasets. And note that the paper requires major revisions in writing. The rebuttal did not adequately address these concerns.

Reviewer aNrj raises concerns about insufficient comparisons with ensemble and knowledge distillation methods. While the rebuttal partially addresses these concerns, the issue of scaling to larger models remains unresolved.

Overall, the experimental analysis in this paper is insufficient for publication.

---

### Decision · Program_Chairs · 2025-01-22

Reject